# Explanation for a necessary change of Eq.4

Unfortuately, we detected a mistake after the review in Eq.4. The 'minus' sign in this equation was included by mistake and went unnoticed during earlier stages by both co-authors and reviewers. This equation represents the standard definition of the mean error and the correct form of the equation does not include a minus sign as given, for example, by Wilks: Statistical Methods in the Atmospheric Sciences, Third Edition, (Eq. 8.32, page 326):

Here $r_{yo}$ is the Pearson product-moment correlation between the forecasts and observations, $s_y$ and $s_o$ are the standard deviations of the marginal distributions of the forecasts and observations, respectively, and the first term in Equation 8.31 is the square of the *mean error*,

$$ME = \frac{1}{n}\sum_{k=1}^{n}(y_k - o_k) = \bar{y} - \bar{o}. \tag{8.32}$$

The mean error is simply the difference between the average forecast and average observation, and therefore expresses the bias of the forecasts. Equation 8.32 differs from Equation 8.30 in that the

individual forecast errors are not squared before they are averaged. Forecasts that are, on average, too large will exhibit ME > 0, and forecasts that are, on average, too small will exhibit ME < 0. It is important to note that the bias gives no information about the typical magnitude of individual forecast errors and is therefore not in itself an accuracy measure.

The correct equation (without the minus sign) has been used to produce the results shown in Fig.4 and the respective figures in the Supplement and the results disscussed in the results and disscusion sections, especially on page 6, lines 53-68. No changes to the results (text or figures) are therefore necessary.

# Statistical calibration of probabilistic medium-range Fire Weather Index CE1 forecasts in Europe

**Stephanie Bohlmann and Marko Laine**

Finnish Meteorological Institute, P.O. Box 503, 00101 Helsinki, Finland

**Correspondence:** Marko Laine (marko.laine@fmi.fi)

**Abstract.** Wildfires are increasing in frequency and severity across Europe, which makes accurate wildfire risk estimation crucial for decision-makers and emergency responders. Wildfire risk is usually estimated using meteorological-based fire weather indices such as the Canadian Forest Fire Weather Index (FWI). By using weather forecasts, the FWI can be predicted for several days and even weeks ahead. Probabilistic ensemble forecasts require verification and calibration in order to provide reliable and accurate forecasts, which are crucial for informed decision-making and an effective emergency response. In this study, we investigate the potential of non-homogeneous Gaussian regression (NGR) for statistically calibrating ensemble forecasts of the FWI. The FWI is calculated using medium-range ensemble forecasts from the European Centre for Medium-Range Weather Forecasts (ECMWF) with lead times up to 15 d over Europe. The method is tested using a 30 d rolling training period and dividing the European region into three training areas (northern, central, and Mediterranean Europe). The calibration improves FWI forecast particularly at shorter lead times up to 84 h and in regions with elevated FWI values, i.e. areas with a higher wildfire risk such as central and Mediterranean Europe. The study demonstrates that NGR can be used to improve probabilistic FWI forecasts especially in the time range most critical for firefighting resource management and thereby supporting effective wildfire response strategies.

## 1 Introduction

Wildfires in Europe have become increasingly frequent and destructive in recent decades. The wildfire season of 2023 alone burned, according to the European Forest Fire Information System (EFFIS, 2024), an area of over $1.7 \times 10^5$ ha in Greece, killed at least 18 people, and forced thousands to leave their homes (Faiola and Labropoulou, 2023). Similarly devastating wildfires in 2017 and 2022 burned large areas in Portugal, Spain, and Italy (Turco et al., 2019; Rodrigues et al., 2023). While the Mediterranean region continues to face the highest occurrence of wildfires, central and northern Europe have experienced an increase in extreme temperature events and heatwaves in recent years (Ibebuchi and Abu, 2023; Rousi et al., 2023; Ionita et al., 2017; Barriopedro et al., 2011). Extended warm and dry periods raise the fire danger and may cause wildfires in regions that were previously not considered wildfire hotspots (San-Miguel-Ayanz et al., 2021; De Rigo et al., 2017). Examples are the 2018 heatwave, which caused wildfires in Sweden (San-Miguel-Ayanz et al., 2019) and across the United Kingdom (Sibley, 2019), and the wildfire outbreaks in Germany and the Czech Republic during the summer of 2022 (Skacel et al., 2022).

The rising frequency and severity of wildfires in Europe emphasize the need for an effective management of forest fire emergencies. The SAFERS (Structured Approaches for Forest fire Emergencies in Resilient Societies, https://safers-project.eu/, last access: 13 November 2024) project provides an integrated platform to assist first responders, firefighters, and decision-makers to become more resilient before, during, and after forest fire emergencies. An important component of SAFERS for identifying high-wildfire-risk areas includes accurate and reliable weather forecasts, ranging from a few days up to 6 weeks. In this paper, we use the Canadian Forest Fire Weather Index (FWI; Van Wagner, 1987; Di Giuseppe et al., 2016), a widely recognized numerical indicator of forest fire risk, to derive fire risk from weather forecasts with a lead time up to 2 weeks. The calculation

of the FWI only requires four weather parameters: temperature, relative humidity, wind speed, and 24 h accumulated precipitation, which are often available from deterministic or probabilistic weather forecasts. While deterministic forecasts provide a single forecast based on a given set of initial conditions, probabilistic ensemble forecasts offer a range of possible outcomes by using slightly perturbed initial conditions, giving a more comprehensive picture of potential weather conditions and providing an estimate of the forecast uncertainty. Probabilistic ensemble forecasts may require statistical calibration to ensure reliable and accurate forecasts, which are essential for making informed decisions and effectively allocating resources when responding to wildfires.

Various methods have been developed for calibrating probabilistic ensemble forecasts. Commonly used calibration methods include Bayesian model averaging (Raftery et al., 2005); non-homogeneous Gaussian regression (Gneiting et al., 2005); logistic regression (Hamill et al., 2004); and non-parametric ensemble post-processing methods such as rank histogram techniques (Hamill and Colucci, 1997), quantile regression (Bremnes, 2004), and ensemble dressing approaches (Roulston and Smith, 2003). Non-homogeneous Gaussian regression (NGR) is one of the most commonly used calibration methods and adjusts both ensemble mean and spread while still being efficient and easy to implement. It has been proven effective for various weather variables like temperature (Hagedorn et al., 2008), precipitation (Hamill et al., 2008), and wind speed (Thorarinsdottir and Johnson, 2012) and can be applied using a truncated or censored distribution to account for a constraint to non-negative values.

In this article, we investigate whether non-homogeneous Gaussian regression (NGR) can be used to calibrate the Fire Weather Index (FWI) derived from medium-range ensemble weather forecasts and assess the extent to which NGR improves the accuracy of FWI predictions. The skill of the calibrated FWI forecast is shown and compared for three European regions.

## 2 Data and methods

### 2.1 Fire Weather Index calculation

A common method for indicating the danger of wildfires is the Canadian Forest Fire Weather Index (FWI) system (Van Wagner, 1987). Although originally developed for Canadian weather and vegetation, the FWI system is now used in various regions (de Groot et al., 2007; Di Giuseppe et al., 2016). For instance, the European Forest Fire Information System (EFFIS) employs the FWI to provide information on wildfires in the EU and neighbouring countries (Di Giuseppe et al., 2020). The FWI system only requires four weather parameters and information about the season (time of year) and geographical location as input parameters and is calculated in two steps: first, the 2 m temperature, 2 m relative humidity,

10 m wind speed, and 24 h accumulated precipitation at local noon are used to calculate the moisture content of three separate fuel layers, i.e. Drought Code (DC), Drought Moisture Code (DMC), and Fine Fuel Moisture Code (FFMC). These fuel layers are characterized by different depths and fuel consistencies, which result in varying water capacities and drying speeds. The Fine Fuel Moisture Code (FFMC) represents the moisture content of litter and other fine cured fuels at a nominal depth of 1.2 cm. The Duff Moisture Code (DMC) indicates the moisture content of loosely compacted layers (nominal depth $\sim$ 7 cm). The Drought Code (DC) denotes the moisture content of deep, compacted layers at a depth of around 18 cm (Van Wagner, 1987). DMC and DC respond slower to weather variations compared to the fast-drying fuel represented by the FFMC. Consequently, the effective day length, which determines the amount of drying that can occur during a given day, must be considered, and monthly day length adjustment factors for DMC and DC are applied based on latitude (Lawson and Armitage, 2008). The fuel moisture codes are dependent on previous weather conditions; therefore, the preceding day's noon values for FFMC, DMC, and DC are necessary for their calculations.

In the second step, FFMC and the 10 m wind speed are used to model the potential rate of fire spread, i.e. Initial Spread Index (ISI). DMC and DC are used to calculate the Buildup Index (BUI), a numeric rating of the total amount of fuel available for combustion, which comprises the potential of a surface fire to burn deeper fuel layers (build-up) and thus evolve into more persistent fires. These fire behaviour indices are then used to calculate the FWI. The FWI values are always non-negative, with low numbers indicating low fire weather danger and high values indicating high fire weather danger. The FWI is often classified into danger classes, with values below 11 considered low and those above 50 classified as extreme, according to the EFFIS fire danger classes (EFFIS, 2020). However, those levels can vary depending on local conditions (e.g. vegetation types). Consequently, what is considered a low or extreme FWI in one region may not be equivalent in another. A more comprehensive description of the FWI system can be found in Van Wagner (1987) and Lawson and Armitage (2008). For this study, we implemented the calculation of the Canadian Forest Fire Weather Index (FWI) using the Python programming language, following the source code provided by Wang et al. (2015).

### 2.2 Forecast and observation data

In this study, we use ensemble forecasts from the European Centre for Medium-Range Weather Forecasts (ECMWF) to calculate ensemble forecasts of the FWI. ECMWF medium-range ensemble forecasts consist of 50 members, initialized twice a day at 00:00 and 12:00 UTC, and provide forecasts up to 360 h (15 d). The forecasts are derived from the TIGGE archive (Bougeault et al., 2010), which provides operational medium-range ensemble forecast data for

non-commercial research purposes. The data are accessible through the ECMWF API (ECMWF, 2024a TS1). The temporal resolution of the ensemble forecast data used in this paper is 6 h for all lead times, and the spatial resolution is 0.5° (∼ 50 km in central Europe). We use only the forecasts initialized at 00:00 UTC. Although available forecasts cover the whole globe, our focus is on the European region from 25 to 72° N and 25° W to 39.80° E, and we specifically use forecasts from the years 2021 to 2023.

For the FWI calculation, we derive initial values for FFMC, DMC, and DC from ERA5 reanalysis data (Hersbach et al., 2020) to account for preceding conditions at forecast initialization. The initial values are determined using the climatological mean from 40 years of historical data (1980–2019) for each day of the year, using a centred 15 d rolling mean on each day. ERA5 reanalysis data can be retrieved from the C3S Climate Data Store (CDS) (Hersbach et al., 2017).

For calibration and verification purposes, we use ECMWF high-resolution deterministic forecasts initialized at 00:00 UTC, which are available from ECMWF's Meteorological Archival and Retrieval System (ECMWF, 2024b TS2). ECMWF high-resolution forecasts have a spatial resolution of 0.1° (∼ 9 km) and a temporal resolution of 1 h and can therefore give a more accurate picture of the actual weather conditions than medium-range ensemble forecasts with a coarser resolution. Ideally, the FWI forecasts would be verified using FWI values calculated from surface observations of the relevant weather parameters, since the FWI cannot be directly observed. However, measurement stations that provide continuous observations of all necessary weather parameters are sparse and only yield pointwise verification. Furthermore, for an operational calibration of the FWI, observation data would need to be rapidly available. We therefore use the FWI calculated using ECMWF high-resolution forecasts with the shortest lead time to the local noon with corresponding 24 h precipitation as a substitute for actual observations. These FWI values are hereafter called the analysis.

To determine whether the analysis is suitable to be used as an observation substitute, we checked their agreement with actual observation-based values, which are shown in Figs. 1 and 2. We use observations available from the Finnish Meteorological Institute's observation database for the years 2021–2023. Figure 1a shows the stations for which it is possible to calculate the FWI for more than 200 consecutive days in addition to the reference areas, which are introduced in Sect. 3. In total 682 stations can be used. The time series of three example stations are shown in Fig. 1b–d. The FWI analysis is shown in orange, while the FWI calculated from observations is shown by the dashed black line. Overall, there is good agreement between the forecasted and observed FWI values. However, especially for high FWI values, the FWI derived from forecasted weather parameters tends to underestimate the values compared to those derived from

observations. This is particularly evident during the summer months of 2022 and 2023 in Meiningen, Germany (Fig. 1c), and Chrysoupoli, Greece (Fig. 1d).

Figure 2a shows overlapping histograms of the density distribution of the FWI analysis (orange) and FWI calculated from observations (grey) for all stations shown in Fig. 1a and every time step. The distributions show a high frequency of very low values (FWI < 1), and when plotted on a logarithmic $x$ scale, a bimodal structure becomes evident with a separation at an FWI value of 1. This bimodality results from a necessary restriction imposed on the FWI calculation (Eq. 30a, b in Wang et al., 2015 or Eq. 40 in Van Wagner, 1987). Figure 2b again displays the density distribution of FWI values but focuses on FWI values greater than 1, as very low values (FWI < 1) can be disregarded when assessing wild fire danger. The analysis tends to overestimate FWI values below 6 and underestimate values above 6. Notably, these higher values (FWI > 10) are the most relevant for assessing wildfire danger. One contributing factor is the different spatial resolution of the data sources. The analysis uses gridded data with a resolution of 0.1° (approximately 9 km in central Europe), whereas weather observations are local measurements. However, in general there is a good positive correlation of FWI analysis and FWI calculated from observations. Figure 2c shows the histogram of the linear correlation coefficient of stations that observe FWI values greater than 1, and the mean correlation coefficient is 0.72. Low correlation coefficients are mainly caused by differences between analysis and observation in mountainous areas, such as Austria, Switzerland, Romania, and Norway, as indicated by the yellow and red coloured markers in the station map (Fig. 1a). These differences result from the difficulty of capturing the complex terrain and its small-scale weather phenomena with the relatively coarse resolution (approximately 9 km) of the forecast model. Notably, 53 % of the available stations are situated in these aforementioned countries, while relatively few stations are located in southern Europe where high FWI values are typically expected. This uneven distribution of observation stations may influence the overall correlation assessment. Moreover, this demonstrates the challenges of using observations for verification and calibration over larger study domains. The spatial distribution of available data across the entire area and the representativeness of pointwise observations for larger regions, particularly in complex terrain, must be considered. Given the generally good correlation between observations and analysis, despite the underestimation of larger FWI values, we will use the forecasted FWI with short lead time (analysis) derived from ECMWF high-resolution forecasts as a proxy for observations in the subsequent evaluations, ensuring the complete spatial coverage across the entire domain.

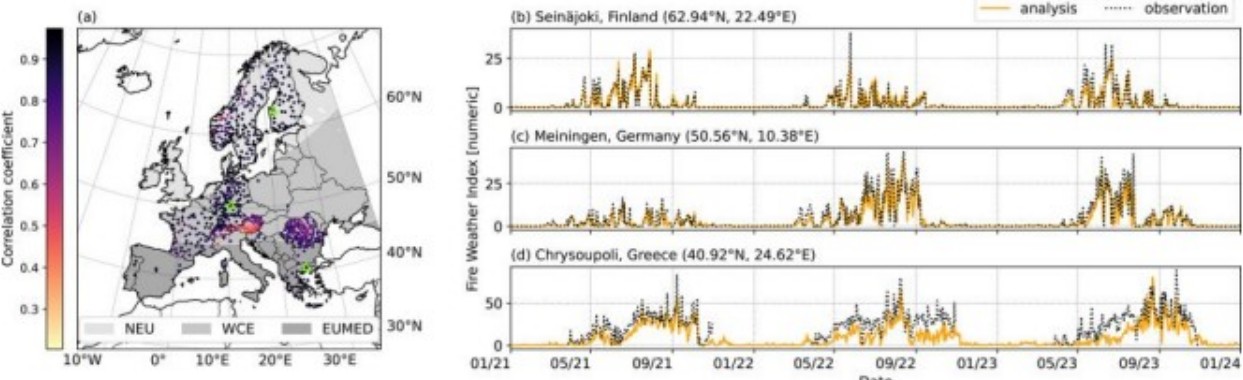

**Figure 1. (a)** Map of study area with modified AR6-WGI reference regions (Iturbide et al., 2020) shaded in grey (NEU – northern Europe, WCE – western and central Europe, EUMED – European Mediterranean). The locations of observation stations for which the FWI can be calculated for at least 200 consecutive days are presented by dots with the colour representing the correlation coefficient (see Fig. 2c) for each station. Example stations **(b–d)** are marked by green triangles. Example of FWI time series calculated from observations (dashed black lines) and forecast analysis (orange) for a station **(b)** in NEU (Seinäjoki, Finland), **(c)** for WCE (Meiningen, Germany), and **(d)** EUMED (Chrysoupoli, Greece).

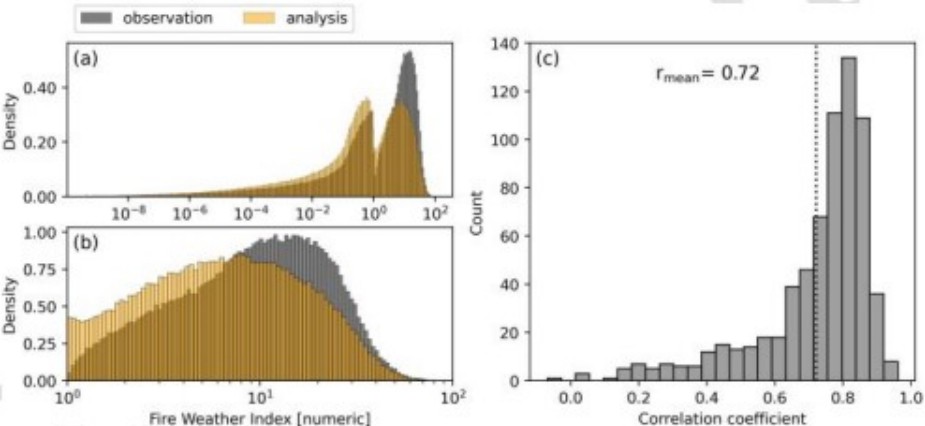

**Figure 2. (a)** Density plot of the FWI calculated from observation data (grey) and from the FWI analysis (orange) for all the stations shown in Fig. 1a. **(b)** Same as in panel **(a)** but only showing FWI values greater than 1. **(c)** Frequency (count) of the correlation coefficient of FWI high-resolution analysis and FWI calculated using observation data for FWI values greater than 1. The vertical line marks the mean correlation coefficient.

## 2.3   Calibration and verification methods

### 2.3.1   Non-homogeneous Gaussian regression

For statistical post-processing the medium-range FWI forecasts, we apply the non-homogeneous Gaussian regression (NGR), which was originally proposed and employed for surface temperature and sea level pressure by Gneiting et al. (2005). This method was extended to non-negative weather variables (wind speed) by Thorarinsdottir and Gneiting (2010). The FWI is by definition non-negative; for the calibration of the FWI forecasts, we assume that the FWI observations $y$ follow a truncated normal distribution with a

cut-off at zero:

$$y \sim \mathcal{N}^0(\mu, \sigma^2). \tag{1}$$

The location and scale parameter are given by

$$\mu_{kl} = a_l + b_l \overline{\mathrm{ens}}_{kl},$$
$$\log(\sigma_{kl}) = c_l + d_l \log(\mathrm{sd}_{kl}), \tag{2}$$

where $\overline{\mathrm{ens}}_{kl}$ is the ensemble mean, and $\mathrm{sd}_{kl}$ is the standard deviation of the 50 ensemble members for each location $k$ and lead time $l$. $a_l$–$d_l$ are regression coefficients. The logarithmic link $\log(\mathrm{sd}_{kl})$ is used to assure positive values for the scale parameter $\sigma_{kl}$.

The regression coefficients $a_l$–$d_l$ are estimated by minimizing the average continuous ranked probability score (CRPS; Hersbach, 2000) from training data. The training data are defined using a specific training area and a 30 d rolling window preceding the forecast as a training period. As training areas, we use here climatic reference regions, defined by the sixth IPCC Assessment Report (AR6-WGI; Iturbide et al., 2020), which divide the European domain into northern Europe (NEU), western and central Europe (WCE), and the Mediterranean (MED). In this study, however, we use only the European part north of the Mediterranean Sea, hereafter referred to as the European Mediterranean (EUMED). The reference regions are marked grey in Fig. 1a. The calibration can also be performed over smaller geographical areas, e.g. individual countries. Training windows from 15 to 40 d were tested, as shorter training periods allow a faster adaptation to seasonal differences. On the other hand, longer periods provide more data, thereby reducing statistical variability. Only minor differences in the calibration performance were found when using big training areas, as in this example. When using smaller geographical training areas, however, a training period of 30 d seemed to be most suitable (see Fig. S1 in the Supplement). Therefore, we chose a 30 d window for our analysis. We adopt here a regional approach, pooling training data from all grid points within the training area to derive a single set of calibration coefficients ($a_l$–$d_l$) specific to each lead time for the given forecast. The initial forecast time step ($T + 12\,\mathrm{h}$) is excluded from calibration, as these forecasts are used as the observation. The obtained coefficients are then used to calibrate the forecast at the respective lead time in the selected training area. Fitting of the regression model and prediction of location and scale parameters of the predicated distribution are done using the R package *crch*, which provides censored regression with conditional heteroscedasticity (Messner et al., 2016) and uses the Broyden–Fletcher–Goldfarb–Shanno algorithm (Nocedal and Wright, 2006) to minimize the CRPS.

### 2.3.2 Verification metrics

The aim of forecast calibration is to correct forecast errors derived from both structural deficiencies in the dynamical models and from CE2 forecast sensitivity to uncertain initial conditions (Wilks and Vannitsem, 2018). To evaluate the predictive performance of calibrated forecasts compared to the raw forecasts, we are using the spread–skill relationship, mean error, and the continuous ranked probability skill score (CRPSS), which are shortly introduced hereafter.

A common method to evaluate the forecast reliability of probabilistic forecasts is the assessment of the spread–skill relationship (Weigel, 2011). A frequently used measure for the skill of an ensemble forecast is the root mean square error (RMSE) of the ensemble mean, calculated as

$$\mathrm{RMSE} = \sqrt{\frac{1}{n}\sum_{i=1}^{n}(F_i - O_i)^2}, \tag{3}$$

where $F_i$ and $O_i$ are the predicted and observed value, respectively, at time step $i$ of $n$ forecasts. The ensemble spread is calculated as the square root of the average ensemble variance, where the average ensemble variance is the mean of the variances of all ensemble members (Fortin et al., 2014). In a well-calibrated forecast model, the ensemble spread should be on average equal to the RMSE. The ensemble forecast is considered underdispersive if the spread is smaller than the RMSE and overdispersive otherwise.

The bias of the forecast can be accessed by simply evaluating the difference between the forecasted value $F_i$ and observed value $O_i$, which is defined as the mean error (ME):

$$\mathrm{ME} = \frac{1}{n}\sum_{i=1}^{n}(F_i - O_i). \tag{4}$$

TS3 While the spread–skill relationship and ME are deterministic scores and applied to the ensemble forecast mean, the continuous ranked probability score (CRPS; Eq. 5) allows a probabilistic assessment (Hersbach, 2000). The CRPS compares the whole distribution of ensemble members, represented as cumulative distribution function, with the observation:

$$\mathrm{CRPS}(P, x_a) = \int_{-\infty}^{\infty} [F_{\mathrm{Fc}}(x) - F_{\mathrm{O}}(x)]^2 \mathrm{d}x, \tag{5}$$

where $F_{\mathrm{Fc}}(x)$ and $F_{\mathrm{O}}(x)$ are the cumulative distribution functions of the forecast and the observation, respectively. The CRPS is negatively oriented, which means smaller values indicate a better performance of the ensemble forecast. In this study, we assume a truncated Gaussian distribution of the FWI forecasts and apply the truncated Gaussian form of the CRPS for raw and calibrated ensemble forecasts. The skill of the calibrated forecast with respect to a reference forecast can be accessed by calculating the continuous ranked probability skill score (CRPSS), defined as

$$\mathrm{CRPSS} = 1 - \frac{\mathrm{CRPS}_{\mathrm{cal}}}{\mathrm{CRPS}_{\mathrm{ref}}}, \tag{6}$$

where $\mathrm{CRPS}_{\mathrm{cal}}$ and $\mathrm{CRPS}_{\mathrm{ref}}$ denote the CRPS of the calibrated and reference forecast, respectively. Positive values indicate a higher skill of the calibrated forecast, while negative values indicate a lower skill of the calibrated forecast with respect to the raw forecast.

## 3   Results

In this section, we present results of post-processing FWI forecasts for the years 2021 to 2023 by applying the NGR method. We use here the AR6-WGI reference regions NEU, WCE, and EUMED introduced in Sect. 2.3.1 and shown in Fig. 1a.

The main fire season in Europe is typically from May until October but varies strongly in length and intensity across regions. For example, the fire season starts later and is shorter in northern Europe compared to southern Europe (San-Miguel-Ayanz et al., 2012). For the calibration verification, we therefore only focus on forecasts during the months of May to October, when the FWI in all regions is substantially higher compared to off-season months (see Fig. 1b–d).

Figure 3 shows the spread–skill relationship for the raw (black) and calibrated (orange) FWI forecast averaged over the grid points within the three study areas NEU (left), WCE (middle), and EUMED (right) and the wildfire season (May to October). Monthly averages can be found in the Supplement (Figs. S4–S6). The RMSE of the climatology, shown by the solid blue line, is calculated similarly to the forecast RMSE. However, it uses the climatology derived from 40 years of ERA5 reanalysis data (see Sect. 2.2) for each day of the year instead of using the forecast. Both calibrated and raw forecasts have a smaller RMSE compared to the climatology, which indicates a general skill of the FWI forecasts compared to the climatology even at longer lead times. For raw and calibrated forecasts the spread (dashed line) is constantly smaller than the RMSE (solid line). This implies that the forecast is underdispersive and lacks spread. However, after calibration the ensemble spread is closer to the RMSE, which indicates that the reliability of the forecast is improved. The calibration also improves forecast accuracy, as indicated by the decreased RMSE, especially during the first forecast days. In northern Europe, the RMSE of the calibrated forecast is slightly above the RMSE of the raw forecast after 180 h (7 d) of forecast, whereas the RMSE of raw and calibrated forecast in central and Mediterranean Europe is similar for forecasts longer than 228 h (9 d) and 300 h (12 d), respectively. The FWI forecast in northern Europe lacks skill for lead times longer than 180 h (7 d), and calibration based on a 30 d rolling window fails to improve and even worsens the forecast. This finding could be explained by the rather small FWI values in NEU and the lower variability of values in this region compared to WCE and EUMED (see Figs. S2 and S3). The applied calibration appears to be more effective for higher FWI values and shows limitations for smaller FWI values, i.e. values < 10. This also explains the regional differences and the better calibration results in the more southern, fire-prone regions with generally higher FWI values compared to northern Europe, where FWI values are often very small.

The mean error (ME) averaged over the three subregions and the fire weather season is shown in Fig. 4. Uncalibrated forecasts have a negative bias for all lead times, indicating that the forecasted FWI values are consistently lower than the observed values. In northern Europe the mean error before calibration is around −0.6 for all lead times. In central and southern Europe, the mean error is more negative but increases with lead time. The improvement in the mean error is especially contributed to forecasts in the months with high FWI, July and August for WCE (Fig. S8), and June to September in EUMED (Fig. S9). After calibration the ME is considerably improved. Best results seem to be achieved in the Mediterranean region, where the mean error after calibration is around zero. In northern and central Europe, the bias is slightly positive after calibration, especially for longer lead times, ranging from less than 0.1 to 0.4.

Figure 5 shows the CRPSS with raw forecasts as a reference. In all three regions, the CRPSS is positive for the first 84 h (3 d) of the forecast, suggesting an improvement in the FWI forecast after calibration for short lead times up to 84 h. However, the lead time up to which the calibration improves the forecasts varies with region. In NEU, the calibration actually worsens the skill of the forecast after 132 h (5 d) of forecast, indicated by a negative median CRPSS. In WCE, the median CRPSS is negative after 204 h (8 d), also indicating a decline in skill for longer lead times. In EUMED, however, the median CRPSS remains positive for all lead times. After 276 h (11 d), the median CRPSS approaches zero, indicating that beyond this point, the calibrated forecast has no skill compared to the raw forecast. The declining skill in NEU and WCE is likely due to lower FWI values in these regions (see Figs. S2 and S3). For low FWI values, particularly those below 10, the calibration becomes less effective. With increasing lead time, the skill worsens especially in mountainous areas in Scandinavia and the Alps and in Atlantic-influenced regions, e.g. northwestern Spain and the United Kingdom (see Fig. 6). In these regions the FWI is generally low throughout the fire season (see Fig. S2).

## 4   Discussion and conclusion

We investigated whether non-homogeneous Gaussian regression (NGR) can be used to calibrate Fire Weather Index (FWI) forecasts based on medium-range ensemble weather forecasts by the European Centre for Medium-Range Weather Forecasts (ECMWF) ensemble forecasts. To estimate the calibration coefficients of the NGR, we employ a truncated Gaussian distribution with a cut-off at zero and use forecast and observation pairs of the last 30 d preceding the forecast. NGR is a well-established and effective method for calibrating probabilistic forecasts and provides good results compared to other calibration approaches, such as bias correction (Cannon, 2018) or correcting the individual input parameters, as described by Worsnop et al. (2021). While correcting individual input parameters can be beneficial, it requires different models for each vari-

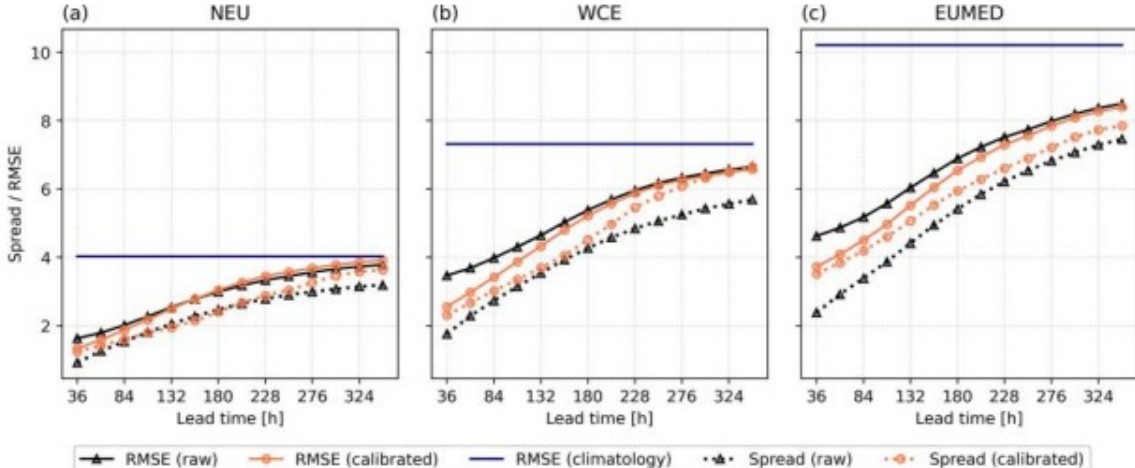

**Figure 3.** Spread and RMSE for raw and calibrated FWI forecasts averaged over the fire season (May–October) and region of interest: **(a)** northern Europe (NEU), **(b)** western and central Europe (WCE), and **(c)** European Mediterranean (EUMED). The RMSE of the climatology for the respective region is additionally provided.

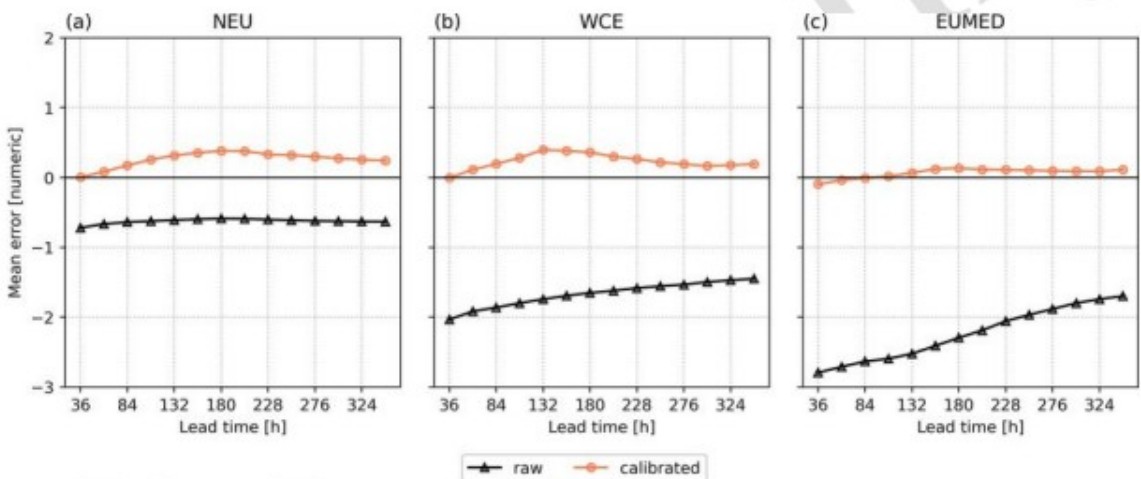

**Figure 4.** Mean error for the raw and calibrated FWI forecasts averaged over the fire season (May–October) and the respective region: **(a)** northern Europe (NEU), **(b)** western and central Europe (WCE), and **(c)** European Mediterranean (EUMED).

able, demanding careful verification and access to quality-controlled observational data over the whole study region. Instead of observed weather parameters, we used ECMWF high-resolution weather forecasts with the shortest possible lead time to calculate the FWI. The FWI values from these forecasts are referred to as the FWI analysis and are used as a substitute for observations. Although the FWI analysis seems to slightly underestimate the FWI in the range associated with elevated fire danger (FWI > 10), a good correlation is observed. Utilizing observations would have been possible but would have required careful consideration of the quality, availability, and representativity of the observation data. Specifically, wind and precipitation observations might not

be of sufficient quality at all stations, and point observations might not accurately represent larger areas. To ensure a consistent coverage of the entire study area and time period, we chose to use the FWI analysis for calibration and verification instead of observations.

FWI forecasts using medium-range ensemble weather forecasts perform generally quite well when compared to the analysis. However, calibration improves the forecasts especially at short lead times up to 84 h (3 d). In the Mediterranean region and central Europe an improvement in FWI forecast with respect to the FWI analysis is also apparent for longer lead times up to 8 to 10 d, respectively. This is likely caused by the generally higher values in those regions

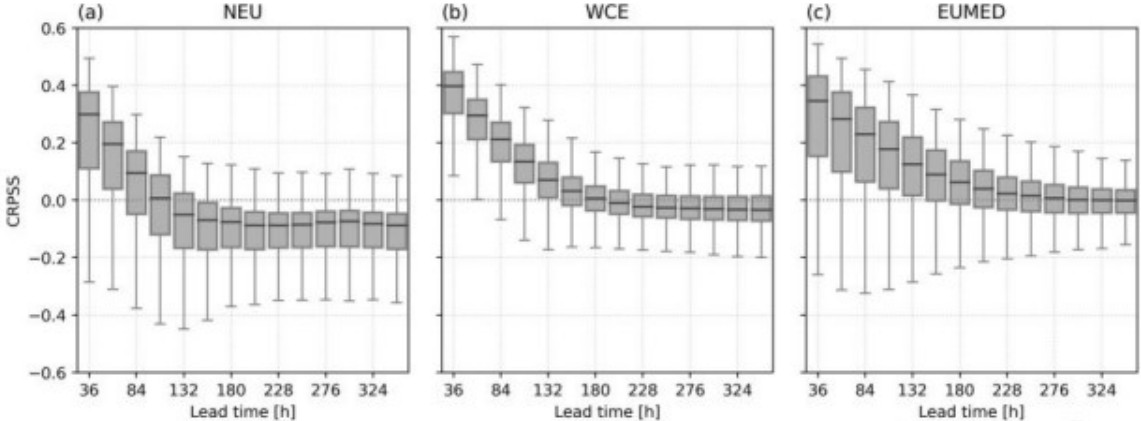

**Figure 5.** Continuous ranked probability skill score (CRPSS) for the calibrated forecasts against raw forecasts averaged over the fire season (May–October) and the respective region: **(a)** northern Europe (NEU), **(b)** western and central Europe (WCE), and **(c)** European Mediterranean (EUMED).

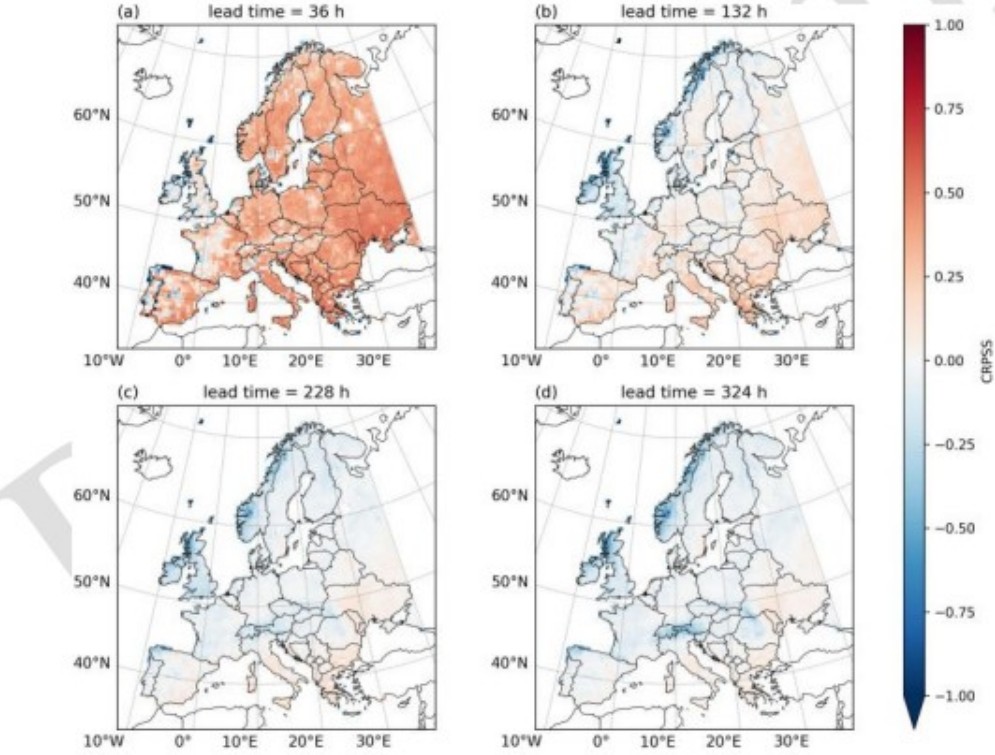

**Figure 6.** Continuous ranked probability skill score (CRPSS) with the raw forecast as a reference averaged over the fire season (May–October) for different lead times.

(shown in Figs. S2 and S3), and further supported by the monthly averaged metrics in the Supplement (Figs. S7–S9), which show a stronger improvement caused by the calibration in the months with high FWI values. Hence, it can be concluded that the calibration performs better for higher FWI values (FWI > 10), as indicated by the better performance during the summer months in western and central Europe and in the European Mediterranean. However, the calibration method shows limitations for low FWI values (FWI < 10), which could be observed in the NEU reference region and especially for longer lead times. Although ideally the method would perform effectively across the entire range of FWI val-

ues, its performance at higher FWI values is generally more critical, as higher values are associated with greater fire danger. Forecasts of potential fire danger for extended periods (1–2 weeks) are valuable; however, short-term forecasts for the first 1–3 d are usually more critical for firefighting resource management.

To further improve the presented calibration method for Fire Weather Index forecasts, it could be tested whether calibration of individual components of the FWI system (e.g. FFMC, DMC, and DC) would improve overall skill of the forecast. Furthermore, more advanced models using additional predictors, e.g. elevation or land use, could improve the calibration but were not tested here.

Reliable and accurate probabilistic forecasts, particularly when the fire risk is high and over the short-term time frame, are crucial for decision-makers and emergency responders to effectively coordinate resources. The improvement in FWI forecasts using the presented calibration method improves the ability to anticipate fire danger, ultimately supporting better response management, and shows that a relatively simple method can provide good results compared to more complex approaches.

*Code and data availability.* The TIGGE data that were used for demonstrating the method are freely available in the TIGGE archive (https://apps.ecmwf.int/datasets/data/tigge/, ECMWF, 2024a). ERA5 reanalysis data that was used to calculate climatologies are freely available in the Climate Data Store (https://doi.org/10.24381/cds.143582cf, Hersbach et al., 2017). Other data and code can be made available by the corresponding author upon request.

*Supplement.* The supplement related to this article is available online at: https://doi.org/10.5194/nhess-24-1-2024-supplement.

*Author contributions.* SB wrote the manuscript with the help of ML. SB and ML developed the FWI calculation and calibration methods.

*Competing interests.* The contact author has declared that none of the authors has any competing interests.

ther geographical representation in this paper. While Copernicus Publications makes every effort to include appropriate place names, the final responsibility lies with the authors.

*Acknowledgements.* The verification is based on TIGGE data. TIGGE (The International Grand Global Ensemble) is an initiative of the World Weather Research Programme (WWRP).

*Financial support.* This research has been supported by the EU Horizon 2020 Structured Approaches for Forest fire Emergencies in Resilient Societies (SAFERS) (grant no. 869353).

*Review statement.* This paper was edited by Yves Bühler and reviewed by two anonymous referees.

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

**Remarks from the language copy-editor**

CE1     This has been capitalized throughout. Please double-check.
CE2     I have added the word "from" here for clarity.

**Remarks from the typesetter**

TS1     Please confirm.
TS2     Please confirm.
TS3     We have not adjusted the equation yet. Meaning and content changes, including changes to values, should be reviewed by the editor before being implemented in the proofreading stage. Please reassess if these changes are strictly necessary before taking this step. For more information, please see our proofreading guidelines at: http://publications.copernicus.org/for_authors/proofreading_guidelines.html. If you want us to change the equation, please prepare an explanatory document (pdf) and highlight the requested change which we can then send to the editor via our system.