# Peer review of "Statistical calibration of probabilistic medium-range fire weather index forecasts in Europe"

_Natural Hazards and Earth System Sciences, 2024_

## Referee Comment (RC1)

**Major comments (substantial doubts to address, which should not require enough extra work to classify the revision as "major")**

**45-47 ". In the second step, FFMC, DMC and DC are used to model the rate of fire spread (ISI) and the potential fuel available for surface fuel consumption (BUI)."** The wind speed is missing (ISI = FFMC + WS10), and it would correlate to the POTENTIAL rate of spread (I mean, it is a variable involving the combustion of surface fuel, like dry leaves and such + the wind). Also, the BUI = DMC + DC, so it is not surface fuel consumption, but it involves the potential for a surface fire to burn the deeper fuel (Build Up) and become a much more persistent fire.

**76-78 "Figure 1b shows the scatter plot of analysis and observations for all stations and every time step. While the FWI derived from the forecasted weather parameters seems to generally underestimate the FWI values compared to the values derived from the observations (slope ∼ 0.63)" AND Figure 1b:** Many doubts arise from this scatterplot:

- The plot itself shows too many points to use a scatter. A density plot NEEDS to be used in this case, or two if you want to show separately the data from Finland.
- The regression which leads to the 0.63 slope seems to be off by looking at the scatterplot, which might be due to the lack of information about the point density. What causes the slope to be 0.63 (and not closer to 1, as the scatterplot would suggest)? Also, please specify the method used for the regression (I assume linear regression).
- I am aware that it is common practice to use the ECMWF analysis at minimal lead time in place of observations, but once you have found an important underestimate like you did, why did you dismiss it so fast? It seems like a very important matter that can have a huge impact on the paper's reliability. Please explain in depth why you can ignore this bias or what you did to correct it.

**78-79 "a correlation is apparent. This good correlation can also be seen in the time series examples for a station in Finland and Greece":** Please provide us with the necessary quantitative information (e.g., correlation coefficients for all the stations) to support this claim, especially coming right after the previous comment. Two sample stations (Figure 1c) are not enough to validate a claim on over 600 others; by this I do not mean that Figure 1c must be removed.

**90-97 "Furthermore, data from all grid points in the training area is used to estimate a single set of coefficients for the given day (regional EMOS)":** some additional explanation is then needed, how do you go from the $\mu_{kl}$ to the estimate used (I guess $\bar{\mu}_l$?)

**Figure 2:**

- It can be made clearer if the legend was more explicit (dotted line with triangles: spread (raw) / solid line with dots: RMSE (calibrated) / etc. )
- The legend, which is relative to all the three graphs must be outside the first graph. Consider also putting everything in a single column.

**152-153 "In Northern Europe, the RMSE of the calibrated forecast is slightly above the RMSE of the raw forecast after 7 days of forecast," :** please provide at least a hypothesis as to why this happens. The sentence on the subsequent lines "The regional differences could be explained with the generally higher FWI values in the more southern, fire prone regions compared to Northern Europe where FWI values are often very small" addresses the regional differences, but the difference from uncalibrated and calibrated NEU RMSE is not addressed.

**Figure 3:**

- The legend, which is relative to all the three graphs must be outside the first graph. Consider also putting everything in a single column.

**Minor comments (typos and formalities)**

**13: "prevalent"**: word choice

**16: "But not"**: cannot start a sentence with "but not"

**18:** unnecessary comma after "periods"

**19 "heatwave 2018":** either heatwave of 2018 or 2018 heatwave

**24** missing (Oxford) comma after "during"

**24-25: "Accurate and reliable weather forecasts ranging from a couple of days to 25 multiple weeks to identify high wildfire risk areas is an important part of SAFERS":** Accurate and reliable weather forecasts, ranging from a couple of days to 25 multiple weeks, are an important part of SAFERS for identifying high wildfire risk areas. (Or equivalent paraphrasis)

**25 "Here,":** "in this paper,"

**26 "short FWI":** (FWI)

**26: "Wagner, 1987":** Author's last name is Van Wagner, throughout the paper

**29: "One widely used"**: A widely used

**36-39 "Although originally developed for Canadian weather and vegetation, it is used in many other regions, e.g., by the European Forest Fire Information System (EFFIS) to provide information on wildfires in the EU and neighboring counties (Giuseppe et al., 2020)"**: sentence needs to be more orderly and written better; also, author's last name is "Di Giuseppe"

**39-40 "One advantage of using FWI is the relatively simple calculation only requiring four weather parameters in addition to information of the season (time of year) and geographical location":** Rephrase, e.g. "The main advantage of using FWI is its relatively simple computation, only requiring four weather parameters and information about the season (time of year) and geographical location"

**42-43 "the moisture content of three separate fuel layers of different depth and diameter":** this is one interpretation of the three parameters (of course, the main one), meaning that -more or less- they contribute to the fire danger with the same time scale of a certain fuel layer. For example, the DC can also be an index of the lack of precipitation for a long time. I tend to be more cautious when interpreting these indices, but it is a relatively small issue.

**48 "Often the FWI is classified":** the FWI is often classified

**50 "e.g. vegetation types"**: since this is the second time it appears, I have to point out that you cannot put "e.g." in the middle of a sentence without it being in parentheses or in a parenthetical expression (between commas). This is not formal enough for a paper, in my opinion.

**52-53 "Fuel moisture codes (FFMC, DMC, DC) and consequently FWI values are dependent on preceding conditions. Thus, the preceding days noon values are used for FWI calculations and the calculations need to be initialized":** To be put above, together with the input variables, and to be written more clearly.

**62 "The resolution of the used TIGGE data":** The resolution of the TIGGE data used in this paper

**65 "can not":** cannot

**Figure 1 (caption):** please provide a reference (IPCC) for the AR5 regions.

**90 Formula (2):** shouldn't it be $\log(\sigma_{kl})$?

**91-92 "The logarithmic link log(sd)":** define sd (ensamble standard deviation?). Besides, isn't it $sd_{kl}$?

**119 "The bias of the forecast can be accessed by simply evaluating the difference between the average forecast and average 120 observation, which is defined as the mean error (ME)":** The bias of the forecast can be accessed by simply evaluating the difference between the average forecast $F_i$ and average observation $O_i$, which is defined as the mean error (ME)

**137 "defined by the 6th IPCC Assessment Report (AR6 (Iturbide et al., 2020))":** did you not show the AR5 regions before (Figure 1a)? It needs to be coherent, at least in the name in the caption of Figure 1a, if the regions did not change.

**140 "Other regions can be selected as well, e.g. the calibration can also be done country-wise or at even smaller level.":** colloquial, rephrase like "the calibration can also be performed over smaller areas (e.g., single countries)"

**158 "the forecasted FWI is too low compared to observations":** rephrase in a more formal "the forecasted FWI underestimates the observations[…]" or similar.

---

## Referee Comment (RC2)

**General text**

The study "statistical calibration of probabilistic medium-range fire weather index forecasts in Europe" of Bohlmann and Leine shows how the Fire Weather Index (FWI) can be calibrated in medium-range weather forecasts to improve FWI in weather forecast and enhance preparedness of fire-fighting resources during high FWI periods. This topic is scientifically important and suits well into the scope of NHESS.

The authors show that their chosen method, i. e., non-homogenous Gaussian regression (NGR) improves the FWI derived from medium-range weather forecasts at shorter lead times by presenting results of different skill metrics, i.e., RSME, ME and CPRSS. I appreciated that the manuscript is well-written and in general easy to follow. Unfortunately, a clear research question is missing, which makes it hard for the reader to know what to expect from the paper. Further, it is not clear in methods and data section for what post-processing steps which datasets are used. This can be improved by revising the manuscript carefully as outlined in the comments below. The results are presented in a clear structure and the figures are easy to interpret, because of the good metric description in the method section. However, the visualization can be improved by minor adjustments. The discussion section is missing, which is unfortunate as a reflection of the authors on the strengths and weaknesses of their method and achieved results, in comparison to other studies would be very valuable for other researchers in this field.

Before publication, the manuscript needs major revisions. I suggest the authors to revise the manuscript carefully, correct and clarify the methods, data and results section and add a discussion section. Please find my specific comments below:

**Major comments**

1. Study workflow is not very clear, e. g. which datasets are used in which step. This is due to missing research questions and the structure of the manuscript. This should be addressed by:
   a. adding research questions, e. g. can NGR be used to calibrate FWI derived from mid-range weather forecasts, how well does NGR improve the FWI derived from mid-range weather forecast. The research questions should be placed at the end of the introduction, as they help the reader to know what to expect from the following chapters.
   b. Restructuring the manuscript by summarizing chapters 2 to 4 to a data and methods section with subchapters (the following is a suggestion):
      i. 1. Introduction, providing research questions at the end of the chapter
      ii. 2. Data and Methods,
         1. 2.1 FWI and FWI calculation
         2. 2.2 Forecast and observation data
         3. 2.3. Validation and Calibration methods
            a. 2.3.1 NGR
            b. 2.3.2 Verification methods
      iii. 3. Results
      iv. 4. Discussion (missing, see other comment.)
      v. 5. Conclusion

2. Introduction: Please add a paragraph why you chose the NGR method over other methods, e. g. other variations of EMOS calibrations or bias-correction methods (see Whan et al. 2021, https://doi.org/10.1016/j.wace.2021.100310).

3. Fire weather index calculation: It is not clear how the FWI is derived. Please clarify in section 2 how you derive the FWI, i. e. which R-package of python package you are using, as there are differences between cffrds and the ECMWF fire products derived from the ECMWF GEFF-Model (see Vitolo et al. 2019, https://doi.org/10.1038/sdata.2019.32)

4. Forecast and observation data:
   a. Lines 57 – 65: You state that you derive the FWI from the ECMWFs operational forecast system (ENS). Later you state that you use the TIGGE dataset. Can you clarify if you used the ENS dataset, the TIGGE dataset or both datasets later in your analysis? I understand that the TIGGE dataset has a higher temporal resolution than the ENS dataset, however, the spatial resolution is coarser (0.5° vs 0.2°). Later in your results section you show the earliest results for 36h lead time. Can you add a sentence why you are more interested in the increased temporal resolution of the TIGGE dataset over the spatial resolution of the ENS dataset in your manuscript?
   b. Lines 65 – 68: These sentences should be moved to the paragraph where you discuss how you verify the FWI from ECMWF data to observation data.
   c. Line 68: "We therefore use the FWI calculated ECMWF high-resolution forecasts …". You did not introduce what the ECMWF high-resolution forecasts are yet. You can optimize this by merging this sentence with the next sentence (i. e. "ECMWF high-resolution forecasts have …"), but it remains now unclear why you introduced the ENS and the TIGGE dataset before. Please clarify in this section on which datasets you derive the FWI from and for which later steps you use which datasets (NGR regression and verification).

5. Results / Figures:
   a. All figures with subplots: add labels for subplots, i. e., (a), (b), (c) as suggested in the NHESS publication guidelines https://www.natural-hazards-and-earth-system-sciences.net/submission.html#figurestables)
   b. In your written text you relate to lead times as days while in the figures you show lead time in hours on the x-axis. Can you synchronize this information? In the current state the reader has to transform between written "7 days" to 7*24h in the x-axis of the respective plot. You could write the hour also in brackets next to the days in the text.

6. Terminology "short lead times": You state multiple times that your results work best for short lead times. Can you clarify in your manuscript how you define short lead times (e. g. 72h or 132h) or be more specific which lead time you still find good performing (e. g. rephrasing to something like: "for short lead times up to 84h").

7. The discussion is missing. In my opinion the discussion is an integral part of a research paper and as the of the study presents a novel way of calibrating mid-range weather forecasts, it would be good to critically reflect on the results:
   a. Is the FWI a suitable predictor for fire events in all three regions? What are the challenges regarding wildfire hazard in the three regions? For example, you could address why the postprocessing works particularly well in the MED and summer

months of WEU and why not for NEU? Further, you could reflect on how low FWI values, as present in NEU, affect your method.

b.  How does your method (NGR) compare to other post-processing methods? Select a two to three different studies, with a similar research question and set your results in a broader context. For example, you could discuss why you are correcting the FWI instead of the input variables of the FWI, why you chose the NGR method and not a bias correction method or machine learning based method (see Whan et al. 2021 https://www.sciencedirect.com/science/article/pii/S2212094721000086) and Worsnop et al. 2021 (https://journals.ametsoc.org/view/journals/wefo/36/6/WAF-D-21-0075.1.xml?alreadyAuthRedirecting)

c.  What can stakeholders take away from your study. You illustrated quite nicely in the introduction that post-processing helps to make accurate forecasts helping first responders. What do you wish this target group takes away from your results, e. g. will more firefighting resources be placed at locations with elevated FWI?

**Minor comments**

1.  Line 6 & Line9: you use the terms post-processing and calibration interchangeable, please choose one term.
2.  Lines 9 – 11: Be more specific about what you mean by short lead times (e. g. 84h?) and regions with elevated FWI (e. g. MEU)
3.  Lines 13: I would drop the word "recent" and replace "wildfire in Greece 2023" by "wildfire season of 2023".
4.  Line 15: Drop "Also" at beginning of sentence.
5.  Line 16: Drop "But" or make this sentence sound more formal.
6.  Line 17: Provide references for your statement.
7.  Line 19: Provide references for your statement.
8.  Line 20: Here it would be great if more than one example could be provided, e. g. one for each subregion.
9.  Lines 22 – 25: Switch the order of the sentence to stress more clearly that weather forecast is part of SAFERS or drop mentioning the project.
10. Line 26: Add Di Giuseppe et al. 2016 (https://doi.org/10.1175/JAMC-D-15-0297.1) as a reference.
11. Line 26: Watch out that your citation tool takes the names correctly. It is van Wagner and Di Giuseppe not Wagner and Giuseppe.
12. End of line 27: Here I am missing a short explanation of what is the difference between deterministic and probabilistic weather forecast. A short explanation would be helpful to emphasize that the topic of the manuscript is postprocessing probabilistic forecasts.
13. Line 28: drop the word "may" or provide a clear statement whether post-processing is needed. Consider also my first comment on your interchangeable usage of post-processing and calibration. This confuses the reader.
14. Line 39: Chose a more scientific formulation than "is a relative simple calculation" for the FWI.
15. Line 43: Add the depth of the moisture levels.
16. Line 48: Rephrase the sentence starting with often, e. g. "The FWI can be classified"
17. Lines 52 – 54:  I would change the order of the sentences to make the statement at the beginning of the paragraph clearer, e. g. "we use climatological mean values …, to account for preceding conditions at the initialization".

18. Lines 59, 63 and 70: Please provide an approximation of the grid resolution in km in brackets?

19. Line 60: it should be "derived from **the** TIGGE archive".

20. Line 61 and following: Please add dataset after TIGGE, i. e., "The TIGGE dataset …".

21. Line 62: Please add "the" to ECMWF API.

22. Line 62: Please rephrase sentence to "the temporal resolution of the TIGGE dataset …".

23. Lines 65 – 78: Please rephrase this paragraph. Keep the statement that the FWI has multiple input variables. Explain for which later steps you use which dataset to calculate the FWI. Mentioning the station data here is confusing.

24. Line 71: I am not sure which dataset you are meaning by "those", please clarify.

25. Line 75: How many of the 682 stations are in Finland and how many are outside of Finland. Can you provide values.

26. Fig 1 / Line 74 (first mentioned):
    o Add letters for subfigures.
    o Fig 1a
        ▪ Use a different projection, e. g. Lambert Conformal Conic, as the northern latitudes are strongly distorted.
        ▪ It would be very nice to have the stations shown in Fig 1c on the map of Fig 1a as well.
        ▪ Place the region legend (i.e., NEU, WCE, EUMED) inside Fig 1a.
        ▪ Place the legend of the countries (i.e., Finland ant others) at a position, where it is clear the legend belongs to both Figures (i. e. Fig 1a and Fig1b)
    o Fig 1b:
        ▪ Provide a legend for the regression line. Is this the line for all stations, or for only "other" stations (outside of Finland) or for only Finland stations?
        ▪ Can you provide a line for Finland as well?
    o Fig 1c:
        ▪ Please show a 3$^{rd}$ station for WCE.
        ▪ Add the location of the stations in Fig 1a.
        ▪ Why are there missing values in the Greece station in the winter of 2022 and 2023? You previously stated that all your selected stations have a sufficient data coverage. Please clarify that your consecutive 200 days refer to the summer half (?) in the Figure caption.

27. Line 77: Provide a reference to Fig 1b as your statements originate from the figure.

28. Line 77 and Fig 1b: Is your, I assume linearly derived correlation, mainly driven by the large number of low FWI values? Fig 1c shows quite apparent that for high FWI values the underestimation is much stronger pronounced than for low FWI values. This would be also a good point to be discussed in the discussion.

29. Line 79: Provide a reference to Fig 1c.

30. Line 80: Please clarify which datasets you use for longer forecasts.

31. Line 83: Please clarify which dataset(s) you mean by the FWI forecasts.

32. Line 86: Please specify what you are calibrating, e. g. the parameters of the NGR or the whole post-processing pipeline.

33. Line 86: It should be: "**the** FWI".

34. Line 91: Please clarify from which dataset the 51 ensemble members come, e. g. by adding (ENS) in brackets. You describe the statistical part very clearly, but it is not clear to which datasets you are applying the formulas.

35. Line 96: Please specify what the training area is.

36. Line 97: Why do you switch terminology from NGR to EMOS, I understand that this is the approach, but it would be good to decide for one name.

37. Lines 98 -100: Can you provide results, e. g. a table or a plot, for these findings. This could be part of your supplementary material.

38. Line 100: Here it becomes apparent, that it is not clear what the training area and hence smaller geographical training areas should be. Please clarify and provide results in the supplementary material.

39. Line 114: Please add a note that you call RMSE later spread and skill metric (i. e. line 122 and line 144).

40. Line 142: please clarify that you compare fire season length of Northern Europe to Southern Europe.

41. Line 145: the grid points "within" rather than "of" the three study areas.

42. Line 146: I can't follow how you derived the RSME of the climatology. Can you describe this briefly.

43. Line 151, Line 163: clarify that your calibration is done as a post-processing.

44. Line 152: Specify what you mean by short lead times, e. g. 132h ?

45. Line 154: Provide lead time in hours in brackets after "7 days", i. e. 7 days (168 h).

46. Figure 2:
    o Add labels (letters a, b, c) to subplots.
    o Place legend outside of NEU to make it clear it belongs to all three subplots.
    o Adjust the y-axis label (Spread/ RMSE) to your figure caption, which is currently "spread and skill". I would expect them to be the same, e. g. spread and RSME in the figure caption or spread and skill in the y-axis label.

47. Line 157: Rather the three subregions than the respective area.

48. Line 158: Rephrase "too low" to something like "lower than observations".

49. Line 160: "The improvement" instead of "This improvement".

50. Line 164: Provide numbers for what you think is slightly positive.

51. Line 167: specify short lead times.

52. Line 166 – 169: This finding would be a good point to discuss in the discussion. For example, I would be interested what these findings imply for the application of your suggested post-processing technique.

53. Figure 3: please add letters to subregions.

54. Figure 4: Please add letters to subregions.

55. Figure 4 caption: drop "the grid of".

56. Lines 171 – 172: Can you discuss this in your discussion section? Does your approach perform well for higher FWI values, suggested by the better performance in WEU in July and August and MED? Does your approach need to perform well on or low no-fire danger days?

57. Figure 5:
    o Add labels to the subplots (i. e., a, b, c, d)
    o Plot the land-sea boundary to improve the visualization.
    o Drop the large white space in the south and west of the plot in such a manner that the plot is filled with results.

58. Line 176: Here you mention the first time that you calibrate the coefficients of the NGR, this is not clear in your previous description of the post-processing method. Please clarify this in the method section. In Line 176 add "of the NGR" after "to estimate the calibration coefficients".

59. Line 177 - 179: This sentence belongs to the discussion section and not the conclusion session. Also, I suggest adding more meaning to this sentence, e. g. what are the implications of sparsely available data?

60. Line 180: Drop "Thus".
61. Line 180: which dataset do you mean by high-resolution weather forecast with short lead time? Is this the third dataset you introduced in the data section?
62. Line 181: Make clear that you mean the dataset "analysis" and not the analysis of the FWI.
63. Line 189: At the end of your conclusion, I would expect a last sentence coming back to your initial statement that this is important for fire resource management and the SAFERS project. Please add such a sentence.
64. Line 209: Add "Di" to "Di Giuseppe".
65. Line 249: Add "Van" to "Van Wagner".

---

## Author Comment (AC1)

**Letter of Reply to Referee 1**

Thank you for reading the manuscript and providing valuable suggestions to improve the paper. Our responses to your comments are shown below in blue, and changes to the manuscript are indicated in italics. Additionally, all modifications are marked in the revised manuscript.

**Major comments**

45-47 ". In the second step, FFMC, DMC and DC are used to model the rate of fire spread (ISI) and the potential fuel available for surface fuel consumption (BUI)." The wind speed is missing (ISI = FFMC + WS10), and it would correlate to the POTENTIAL rate of spread (I mean, it is a variable involving the combustion of surface fuel, like dry leaves and such + the wind). Also, the BUI = DMC + DC, so it is not surface fuel consumption, but it involves the potential for a surface fire to burn the deeper fuel (Build Up) and become a much more persistent fire.

Thank you for your comment. The suggestions have been included in the revised manuscript. The description of the FWI calculation reads now as follows:

*"The FWI is calculated in two steps. First, the 2-meter temperature, 2-meter relative humidity, 10-meter wind speed, and 24h accumulated precipitation at local noon are used to calculate the moisture content of three separate fuel layers. These fuel layers are characterized by different depths and fuel consistencies, which result in varying water capacities and drying speeds. The Fine Fuel Moisture Code (FFMC) represents the moisture content of litter and other fine cured fuels at a nominal depth of 1.2 cm; the Duff Moisture Code (DMC) indicates the moisture content of loosely compacted layers (nominal depth ~7 cm); and the Drought Code (DC) denotes the moisture content of deep, compacted layers in a depth of around 18 cm (Van Wagner, 1987). DMC and DC respond slower to weather variations compared to the fast-drying fuel represented by the FFMC. Consequently, the effective day length, which determines the amount of drying that can occur during a given day, must be considered and monthly day length adjustment factors for DMC and DC are applied based on latitude (Lawson and Armitage, 2008). The fuel moisture codes are dependent on previous weather conditions; therefore, the preceding day's noon values for FFMC, DMC and DC are necessary for their calculations. In the second step, FFMC and the 10-meter wind speed are used to model the potential rate of fire spread (ISI). DMC and DC are used to calculate the Buildup Index (BUI), a numeric rating of the total amount of fuel available for combustion which comprises the potential of a surface fire to burn deeper fuel layers (build up) and thus evolve into more persistent fires. These fire behaviour indices are then used to calculate the FWI. The FWI values are always non-negative, with low numbers indicating low fire weather danger and high values indicating high fire weather danger. The FWI is often classified into danger classes and values above 50 are considered extreme. However, those levels can vary depending on local conditions (e.g., vegetation types). Consequently, what is considered a low or extreme FWI in one region may not be equivalent in another. A more comprehensive description of the FWI system can be found in Van*

*Wagner (1987) and Lawson and Armitage (2008). For this study, we implemented the calculation of the Canadian Forest Fire Weather Index (FWI) using Python programming language, following the source code provided by Wang et al. (2015), with modifications to utilize gridded input data."*

76-78 "Figure 1b shows the scatter plot of analysis and observations for all stations and every time step. While the FWI derived from the forecasted weather parameters seems to generally underestimate the FWI values compared to the values derived from the observations (slope ~ 0.63)" AND Figure 1b: Many doubts arise from this scatterplot:

- The plot itself shows too many points to use a scatter. A density plot NEEDS to be used in this case, or two if you want to show separately the data from Finland.

Thank you for your comment. We agree that the scatterplot may not have been the most suitable choice for representing our data and conveying our message. We decided to change the representation to histograms showing the distribution of the FWI calculated from high-resolution forecast (analysis) and from observation (Fig.2 in the revised manuscript). Additionally, we split the Figure into 2 parts to avoid overcrowding the Figure. The new Fig.1 contains the map of observation stations (a), with colour representing the linear correlation coefficient, and the time series of three example stations (b,c,d). The new Figure 2 contains the histograms of the frequency distribution of the FWIs (a,b) and the linear correlation coefficient.

- The regression which leads to the 0.63 slope seems to be off by looking at the scatterplot, which might be due to the lack of information about the point density. What causes the slope to be 0.63 (and not closer to 1, as the scatterplot would suggest)? Also, please specify the method used for the regression (I assume linear regression).

The regression line in the original Figure 1 appeared off due to the high concentration of values near zero. This becomes now evident considering the distribution shown in histogram Fig.2a. We also clarified in the text that we used linear regression to obtain the coefficient.

- I am aware that it is common practice to use the ECMWF analysis at minimal lead time in place of observations, but once you have found an important underestimate like you did, why did you dismiss it so fast? It seems like a very important matter that can have a huge impact on the paper's reliability. Please explain in depth why you can ignore this bias or what you did to correct it.

We agree that a correction of the analysis based on the observations would be beneficial. We found that the strongest discrepancies between observation and analysis occur in mountainous areas, e.g., Austria or Norway. This is likely caused by difficulties in capturing the small-scale weather phenomena occurring the complex terrain with relatively coarse resolution of the forecast model. However, in those regions the fire danger is generally rather small because of high precipitation and low temperatures. To further investigate this bias, additional station-specific characteristics such as elevation or land use would be

necessary. In this article, we want to present an easy, straightforward method to calibrate the FWI and we believe that the correlation of analysis and observation is already quite good and justifies the use of the analysis as substitute for observations. However, the use of additional station-specific parameters could be topic of future research.

The section that previously related to the scatterplot has been changed to:

*"To determine if the analysis is suitable to be used as observation substitute, we check their agreement with actual observation-based values, which is shown in Fig. observation and 2. We use observations available from the Finnish Meteorological Institute's observation database for the years 2021–2023. Figure 1a shows the stations, for which it is possible to calculate the FWI for more than 200 consecutive days in addition to the reference areas which are introduced in Chapter 3. In total 682 stations can be used. The time series of three example stations are shown in Fig.1a,b,c. The FWI analysis is shown in orange while the FWI calculated from observations is shown by the black dashed line. Overall, there is good correlation between the forecasted and observed FWI values. However, especially for high FWI values, the FWI derived from forecasted weather parameters tends to underestimate the values compared to those derived from observations. This is particularly evident during the summer months of 2022 and 2023 in Meiningen, Germany (Fig.1c) and Chrysopouli, Greece (Fig.1d).*

*Figure 2a overlapping histograms of the density distribution of the FWI analysis (orange) and FWI calculated from observations (grey) for all stations shown in Fig.1a and every time step. The distributions show a high frequency of low values (FWI< 1), and when plotted on a logarithmic x-scale, a bimodal structure becomes evident with a separation at an FWI value of 1. This bi-modality results from a necessary restriction imposed on the FWI calculation (Eq. 30a,b Wang et al. (2015) or Eq.40 in Van Wagner (1987)). Figure 2b again displays the density distribution of FWI values, but focuses on FWI values greater than 1, as these are generally more relevant for assessing wildfire danger. For higher FWI values, the forecast-derived FWI underestimates those calculated from observations, while for lower FWI values, it tends to overestimate. One contributing factor is the different spatial resolution of the data sources. The analysis uses gridded data with a resolution of 0.1° (approximately 9 km in Central Europe), whereas weather observations are local measurements. However, in general there is a fairly good positive correlation of FWI analysis and FWI calculated from observations. Figure 2c shows the histogram of the linear correlation coefficient of stations that observe FWI values greater than 1, the mean correlation coefficient is 0.72. Low correlation coefficients are mainly caused by differences between analysis and observation in mountainous areas, such as Austria, Switzerland, Romania, and Norway, as indicated by the colored markers in the station map (Fig. 1a). This is due to the difficulty in capturing the complex terrain and its small-scale weather phenomena with relatively coarse resolution (~9km) of the forecast model. Because there is a fairly good correlation of observation and analysis, we will use the forecasted FWI with short lead time (analysis) derived from ECMWF high-resolution forecasts as observation to compare with the FWI forecasts derived from ECMWF medium-range ensemble forecasts."*

[Figure]

Fig.1

[Figure]

Fig.2

78-79 "a correlation is apparent. This good correlation can also be seen in the time series examples for a station in Finland and Greece": Please provide us with the necessary quantitative information (e.g., correlation coefficients for all the stations) to support this claim, especially coming right after the previous comment. Two sample stations (Figure 1c) are not enough to validate a claim on over 600 others; by this I do not mean that Figure 1c must be removed.

We agree that only showing two examples is not enough to show that there is a good correlation. In the revised manuscript we included an additional figure (Fig.2c) which shows the distribution of the correlation coefficient for all stations. The mean correlation coefficient is 0.72, which indicates a fairly good positive relationship correlation. Additionally, we added a third station in Fig.1, providing an example for each subregion. Changes to the manuscript can be seen in the reply to the previous comment.

90-97 "Furthermore, data from all grid points in the training area is used to estimate a single set of coefficients for the given day (regional EMOS)": some additional explanation is then needed, how do you go from the $\mu_{kl}$ to the estimate used (I guess $\mu_l$ ?)

We improved this section (now 2.3.1) to make the method better understandable and to clarify what training data is used. Specifically, we changed this sentence to:

*"We adopt here a regional approach, pooling training data from all grid points within the training area to derive a single set of calibration coefficients ($a_l - d_l$) specific to each lead time for the given forecast."*

We hope this clarifies that the calibration coefficients are estimated by the calibration method and then used to calibrate the ensemble mean of the given forecast.

Figure 2:

- It can be made clearer if the legend was more explicit (dotted line with triangles: spread (raw) / solid line with dots: RMSE (calibrated) / etc. )

We made the legend more explicit as suggested.

- The legend, which is relative to all the three graphs must be outside the first graph. Consider also putting everything in a single column.

We placed the legend outside as suggested.

The previous Figure 2 is now Figure 3 in the revised manuscript:

[Figure]

*Fig.3*

**152-153 "In Northern Europe, the RMSE of the calibrated forecast is slightly above the RMSE of the raw forecast after 7 days of forecast," : please provide at least a hypothesis as to why this happens. The sentence on the subsequent lines "The regional differences could be explained with the generally higher FWI values in the more southern, fire prone regions compared to Northern Europe where FWI values are often very small" addresses the regional differences, but the difference from uncalibrated and calibrated NEU RMSE is not addressed.**

We added a hypothesis trying to explain the slightly decreasing RMSE of the calibrated forecasts in NEU as follows:

*"In Northern Europe, the RMSE of the calibrated forecast is slightly above the RMSE of the raw forecast after 7 days (180 hours) of forecast, whereas the skill of raw and calibrated forecast in Central and Mediterranean Europe is similar for forecasts longer than 9 days (228 hours) and 12 days (300 hours), respectively. The FWI forecast in Northern Europe lacks skill for lead times longer than 7 days and calibration based on 30 day rolling window fails to improve and even worsens the forecast. This finding could be explained by the rather small FWI values and large relative uncertainty in NEU, unlike FWI values in WCE and EUMED (not shown here). The applied calibration appears to be more effective for higher FWI values and shows limitations for smaller FWI values close to 0. This also explains the regional differences and the better calibration results in the more southern, fire prone regions with generally higher FWI values compared to Northern Europe, where FWI values are often very small."*

**Figure 3:**

- The legend, which is relative to all the three graphs must be outside the first graph. Consider also putting everything in a single column.

The legend has been placed below the figure to make it clear that it relates to all three subplots. Figure 3 is now numbered as Fig.4:

[Figure]

*Fig.4*

**Minor comments (typos and formalities)**

13: "prevalent": word choice

We changed the wording to "*frequent*".

16: "But not": cannot start a sentence with "but not"

*We rephrased this sentence as follows:*

*"While the Mediterranean region continues to face the highest occurrence of wildfires, Central and Northern Europe have experienced an increase in extreme temperature events and heatwaves in recent years (Ibebuchi and Abu, 2023; Rousi et al., 2023; Ionita et al., 2017; Barriopedro et al., 2011)."*

*Furthermore, we added references to support the statement.*

18: unnecessary comma after "periods"

*The unnecessary comma has been removed.*

19 "heatwave 2018": either heatwave of 2018 or 2018 heatwave

*We changed it to "the 2018 heatwave".*

24 missing (Oxford) comma after "during"

*The comma has been added.*

24-25: "Accurate and reliable weather forecasts ranging from a couple of days to multiple weeks to identify high wildfire risk areas is an important part of SAFERS": Accurate and reliable weather forecasts, ranging from a couple of days to multiple weeks, are an important part of SAFERS for identifying high wildfire risk areas. (Or equivalent paraphrasis)

*We rephrased this sentence and changed the order of the sentence to make it more clear that the weather forecasts are part of SAFERS.*

*"An important component of SAFERS for identifying high wildfire risk areas are accurate and reliable weather forecasts, ranging from a few days to several weeks."*

25 "Here,": "in this paper,"

*Wording has been changed to "in this paper"*

26 "short FWI": (FWI)

*This has been changed.*

26: "Wagner, 1987": Author's last name is Van Wagner, throughout the paper

*Thank you for pointing this out. This was a mistake in the used Bibtex reference and has been fixed.*

29: "One widely used": A widely used

A paragraph mentioning other calibration methods has been added to the introduction following the suggestion of referee 2 (comment 2). The wording has therefore changed:

*"Various methods have been developed for calibrating probabilistic ensemble forecasts. Commonly used calibration methods include Bayesian model averaging (Raftery et al., 2005), non-homogeneous Gaussian regression (Gneiting et al., 2005), logistic regression (Hamill et al., 2004) and non-parametric ensemble post-processing methods such as rank histogram techniques (Hamill and Colucci, 1997), quantile regression (Bremnes, 2004) and ensemble dressing approaches (Roulston and Smith, 2003). Non-homogeneous Gaussian regression (NGR) is one of the most commonly used calibration methods and adjusts both ensemble mean and spread, while still be efficient and easy to implement. It has been proved effective for various weather variables like temperature (Hagedorn et al., 2008), precipitation (Hamill et al., 2008) or wind-speed (Thorarinsdottir and Johnson, 2012) and can be applied using a truncated or censored distribution to account for a constraint to non-negative values."*

36-39 "Although originally developed for Canadian weather and vegetation, it is used in many other regions, e.g., by the European Forest Fire Information System (EFFIS) to provide information on wildfires in the EU and neighboring counties (Giuseppe et al., 2020)": sentence needs to be more orderly and written better; also, author's last name is "Di Giuseppe"

We improved this sentence in the revised manuscript and corrected the author's name.

*"Although originally developed for Canadian weather and vegetation, the FWI system is now used in various regions. For instance, the European Forest Fire Information System (EFFIS) employs the FWI to provide information on wildfires in the EU and neighboring countries (Di Giuseppe et al., 2020)."*

39-40 "One advantage of using FWI is the relatively simple calculation only requiring four weather parameters in addition to information of the season (time of year) and geographical location": Rephrase, e.g. "The main advantage of using FWI is its relatively simple computation, only requiring four weather parameters and information about the season (time of year) and geographical location"

We rephrased this sentence to:

*"The main advantage of the FWI system is its relatively straightforward computation, requiring only four weather parameters and information about the season (time of year) and geographical location as input parameters."*

42-43 "the moisture content of three separate fuel layers of different depth and diameter": this is one interpretation of the three parameters (of course, the main one), meaning that - more or less- they contribute to the fire danger with the same time scale of a certain fuel layer. For example, the DC can also be an index of the lack of precipitation for a long time. I tend to be more cautious when interpreting these indices, but it is a relatively small issue.

We rephrased this section to add the depth of the moisture levels (see comment 15, referee 2). We also added a comment about the response of the moisture codes to weather variations, please see reply to the first comment.

48 "Often the FWI is classified": the FWI is often classified

We changed the order of this sentence accordingly.

50 "e.g. vegetation types": since this is the second time it appears, I have to point out that you cannot put "e.g." in the middle of a sentence without it being in parentheses or in a parenthetical expression (between commas). This is not formal enough for a paper, in my opinion.

We corrected this throughout the paper. Thank you for this comment!

52-53 "Fuel moisture codes (FFMC, DMC, DC) and consequently FWI values are dependent on preceding conditions. Thus, the preceding days noon values are used for FWI calculations and the calculations need to be initialized": To be put above, together with the input variables, and to be written more clearly.

We moved the sentence about the initial values to the paragraph about the moisture code calculation (see reply to first comment). The sentence about the dataset used for initial values was moved to the Forecast and observation data section (2.2).

62 "The resolution of the used TIGGE data": The resolution of the TIGGE data used in this paper

We rephrased this sentence accordingly.

65 "can not": cannot

This typo has been fixed.

Figure 1 (caption): please provide a reference (IPCC) for the AR5 regions.

We wrongfully called the regions here AR5 when it was supposed to be the AR6-WGI regions. We corrected that and added the reference for the AR6 regions.

90 Formula (2): shouldn't it be $\log(\sigma_{kl})$?

It should indeed be $\log(\sigma_{kl})$. We apologize for this mistake and correct the formula.

91-92 "The logarithmic link log(sd)": define sd (ensemble standard deviation?). Besides, isn't it $sd_{kl}$ ?

sd is defined as the standard deviation of the 50 ensemble members in the previous sentence. We agree with the reviewer that it should be $sd_{kl}$ and corrected this mistake.

119 "The bias of the forecast can be accessed by simply evaluating the difference between the average forecast and average observation, which is defined as the mean error (ME)":

The bias of the forecast can be accessed by simply evaluating the difference between the average forecast $F i$ and average observation $O i$, which is defined as the mean error (ME)

We added the symbols $F_i$ and $O_i$ to the sentence.

137 "defined by the 6th IPCC Assessment Report (AR6 (Iturbide et al., 2020))": did you not show the AR5 regions before (Figure 1a)? It needs to be coherent, at least in the name in the caption of Figure 1a, if the regions did not change.

Thank you for making us aware of this mistake. The reference regions did not change and it is supposed to be AR6. We changed it to be coherent throughout the manuscript.

140 "Other regions can be selected as well, e.g. the calibration can also be done country-wise or at even smaller level.": colloquial, rephrase like "the calibration can also be performed over smaller areas (e.g., single countries)"

We rephrased this sentence to:

*"The calibration can also be performed over smaller geographical areas, e.g., individual countries."*

158 "the forecasted FWI is too low compared to observations": rephrase in a more formal "the forecasted FWI underestimates the observations[…]" or similar.

We rephrased the sentence to:

*"Uncalibrated forecasts have a negative bias for all lead times, indicating that the forecasted FWI values are consistently lower than the observed values."*

---

## Author Comment (AC3)

**Letter of Reply to Referee 2**

Thank you for reading the manuscript and providing valuable suggestions to improve the paper. Our responses to your comments are shown below in blue, and changes to the manuscript are indicated in italics. Additionally, all modifications are marked in the revised manuscript.

**Major comments**

1. Study workflow is not very clear, e. g. which datasets are used in which step. This is due to missing research questions and the structure of the manuscript. This should be addressed by:

    a. adding research questions, e. g. can NGR be used to calibrate FWI derived from mid-range weather forecasts, how well does NGR improve the FWI derived from mid-range weather forecast. The research questions should be placed at the end of the introduction, as they help the reader to know what to expect from the following chapters.

Thank you for this comment. We added a sentence stating the research question at the end of the introduction, as follows:

*"In this article, we investigate whether non-homogeneous Gaussian regression (NGR) can be used to calibrate the Fire Weather Index (FWI) derived from medium-range ensemble weather forecasts and assess the extent to which NGR improves the accuracy of the FWI predictions."*

    b. Restructuring the manuscript by summarizing chapters 2 to 4 to a data and methods section with subchapters (the following is a suggestion):
          i. 1. Introduction, providing research questions at the end of the chapter
          ii. 2. Data and Methods,
               1. 2.1 FWI and FWI calculation
               2. 2.2 Forecast and observation data
               3. 2.3. Validation and Calibration methods
                    a. 2.3.1 NGR
                    b. 2.3.2 Verification methods
          iii. 3. Results
          iv. 4. Discussion (missing, see other comment.)
          v. 5. Conclusion

We agree that the restructuring of the manuscript makes the workflow better understandable and adjusted the structure of the manuscript. However, we decided to integrate the Discussion into the conclusion as "4. Discussion and Conclusion". The new structure of the revised manuscript is as follows:

2. Introduction: Please add a paragraph why you chose the NGR method over other methods, e.g. other variations of EMOS calibrations or bias-correction methods (see Whan et al. 2021, https://doi.org/10.1016/j.wace.2021.100310).

We added a short paragraph at the end of the introduction mentioning other calibration methods and why we chose NGR.

*"Various methods have been developed for calibrating probabilistic ensemble forecasts. Commonly used calibration methods include Bayesian model averaging (Raftery et al., 2005), non-homogeneous Gaussian regression (Gneiting et al., 2005), logistic regression (Hamill et al., 2004) and non-parametric ensemble post-processing methods such as rank histogram techniques (Hamill and Colucci, 1997), quantile regression (Bremnes, 2004) and ensemble dressing approaches (Roulston and Smith,2003). Non-homogeneous Gaussian regression (NGR) is one of the most commonly used calibration methods and adjusts both ensemble mean and spread, while still be efficient and easy to implement. It has been proved effective for various weather variables like temperature (Hagedorn et al., 2008), precipitation (Hamill et al., 2008) or wind-speed (Thorarinsdottir and Johnson, 2012) and can be applied using a truncated or censored distribution to account for a constraint to non-negative values."*

3. Fire weather index calculation: It is not clear how the FWI is derived. Please clarify in section 2 how you derive the FWI, i.e. which R-package or python package you are using, as there are differences between cffrds and the ECMWF fire products derived from the ECMWF GEFF- Model (see Vitolo et al. 2019, https://doi.org/10.1038/sdata.2019.32)

We agree that it was not very clear how we calculate the FWI. We added a sentence in the end of section 2.1 stating that we have implemented the FWI calculation use Python programming language and following the source code provided by Wang et al. 2015, (https://publications.gc.ca/site/eng/9.805607/publication.html), as follows:

"For this study, we implemented the calculation of the Canadian Forest Fire Weather Index (FWI) using Python programming language, following the source code provided by Wang et al. (2015), with modifications to utilize gridded input data."

To check if our implementation provides consistent values with other FWI algorithms, we tested our method using the test values given in Vitolo et al. 2019 and default initial values for FFMC of 85, DMC of 6 and DC of 15 (instead of the ERA5 climatology values used in the presented paper). The outputs closely match those of ECMWF fire products and cffrds, either exactly matching cffrds outputs or falling between the values of ECMWF fire products and cffrds

| Code Source | FFMC | DMC | DC | ISI | BUI | FWI |
|---|---|---|---|---|---|---|
| R package cffdrs * | 87.69 | 7.29 | 17.76 | 10.85 | 7.28 | 9.46 |
| ECMWF * | 87.70 | 8.54 | 19.01 | 10.80 | 8.49 | 10.10 |
| This study | 87.69 | 7.59 | 18.76 | 10.85 | 7.59 | 9.62 |

* values taken from Vitolo et al. 2019

4. Forecast and observation data:

   a.  Lines 57 – 65: You state that you derive the FWI from the ECMWFs operational forecast system (ENS). Later you state that you use the TIGGE dataset. Can you clarify if you used the ENS dataset, the TIGGE dataset or both datasets later in your analysis? I understand that the TIGGE dataset has a higher temporal resolution than the ENS dataset, however, the spatial resolution is coarser (0.5° vs 0.2°). Later in your results section you show the earliest results for 36h lead time. Can you add a sentence why you are more interested in the increased temporal resolution of the TIGGE dataset over the spatial resolution of the ENS dataset in your manuscript?

The mentioning of ENS and TIGGE dataset is indeed confusing. In this study, we use the ECMWF ensemble forecasts retrieved from the TIGGE database (spatial resolution of 0.5°), although the method is intended for post-processing the operational ECMWF forecast (spatial resolution of 0.2°) to provide FWI forecasts for the SAFERS project. However, we don't have access to multiple years of ECMWF ENS forecast data with higher spatial resolution. To still perform a statistical analysis, we decided to use TIGGE data instead.

To avoid confusion, we decided not to mention ECMWF operational forecasts. Instead, we introduce in the beginning of the chapter 2.2 "Forecast and observation data" all datasets that are used in this study, i.e., the TIGGE ECMWF forecasts, ERA5 reanalysis and High-resolution forecasts.

*"2.2 Forecast and observation data*

*In this study, we use ensemble forecasts from the European Centre for Medium-Range Weather Forecasts (ECMWF) to calculate ensemble forecasts of the FWI. ECMWF medium-range ensemble forecasts consists of 50-members, initialized twice a day at 0000 and 1200*

*UTC and provide forecasts up to 360 hours (15 days). The forecasts are derived from the TIGGE archive (Bougeault et al., 2010), which provides operational medium-range ensemble forecast data for non-commercial research purposes. The data is accessible through the ECMWF API1. The temporal resolution of the ensemble forecast data used in this paper is 6h for all lead times and the spatial resolution is 0.5° (~50 km). We use only the forecasts initialized at 0000 UTC. Although available forecasts cover the whole globe, our focus is on the European region from 25°N to 72°N and 25°W to 39.80°E and we specifically use forecasts from the years 2021 to 2023.*

*For the FWI calculation, we derive initial values for FFMC, DMC, and DC from ERA5 reanalysis data (Hersbach et al., 2020) to account for preceding conditions at forecast initialization. The initial values are determined using the climatological from 40 years of historical data (1980-2019) for each day of the year, using a 15-day rolling mean centered on each day. ERA5 reanalysis data can be retrieved from the C3S Climate Data Store (CDS) (Hersbach et al., 2017).*

*For calibration and verification purposes, we use ECMWF high-resolution forecasts initialized at 0000 UTC. ECMWF high-resolution forecasts have a spatial resolution of 0.1° (~9 km) and a temporal resolution of 1 hour and can therefore give a more accurate picture of the actual weather conditions than medium-range ensemble forecasts with a coarser resolution. The FWI values calculated using ECMWF high-resolution forecasts with the shortest lead time to the local noon with corresponding 24h precipitation act as substitute for actual observations and are hereafter called analysis. Ideally, FWI forecasts would be verified using FWI values calculated using surface observations of the relevant weather parameters as the FWI cannot be observed directly. However, measurement stations that provide continuous observations of all necessary weather parameters are sparse and only yield point-wise verification. Furthermore, for an operational calibration of the FWI, observation data needs to be available rapidly."*

b. Lines 65 – 68: These sentences should be moved to the paragraph where you discuss how you verify the FWI from ECMWF data to observation data.

We moved these sentences to fit better in the context of the verification of FWI analysis and observation data. Please see changes to Section 2.2 given in the previous comment.

c. Line 68: "We therefore use the FWI calculated ECMWF high-resolution forecasts …". You did not introduce what the ECMWF high-resolution forecasts are yet. You can optimize this by merging this sentence with the next sentence (i. e. "ECMWF high-resolution forecasts have …"), but it remains now unclear why you introduced the

ENS and the TIGGE dataset before. Please clarify in this section on which datasets you derive the FWI from and for which later steps you use which datasets (NGR regression and verification).

This section (2.2) has been completely rewritten to make it more clear which dataset is used for which step and to accommodate the changes to Figure 1 suggested by referee 1. Please see reply to comment 4a and second major comment of referee 1.

5. Results / Figures:

   a.   All figures with subplots: add labels for subplots, i. e., (a), (b), (c) as suggested in the NHESS publication guidelines https://www.natural-hazards-and-earth-system-sciences.net/submission.html#figurestables)

Labels have been added to all subplots to be in accordance with the guidelines.

   b.   In your written text you relate to lead times as days while in the figures you show lead time in hours on the x-axis. Can you synchronize this information? In the current state the reader has to transform between written "7 days" to 7*24h in the x-axis of the respective plot. You could write the hour also in brackets next to the days in the text.

We believe the article is easier to follow using lead times in days instead of hours but agree that using both units throughout the manuscript is confusing. We applied the suggested changes and added the lead time in hours after every mention of the lead time in days. Please note that the lead times in hours do not exactly correspond to the typical conversion of days to hours (e.g., 7 days is here 180 hours instead of 168 hours). This is because FWI forecast are issued for 12 UTC while they are initialized at 00 UTC. The first day's forecast has therefore a lead time of 36 hours and not 24 hours.

6. Terminology "short lead times": You state multiple times that your results work best for short lead times. Can you clarify in your manuscript how you define short lead times (e. g. 72h or 132h) or be more specific which lead time you still find good performing (e. g. rephrasing to something like: "for short lead times up to 84h").

Thank you for this comment. We agree that it is not very clear what is meant with short lead times. We clarified in the manuscript that an improvement is visible for all regions up to a lead time of 84 hours, and we added this information to every instance where we mention that the method works best for short lead times.

7. The discussion is missing. In my opinion the discussion is an integral part of a research paper and as the of the study presents a novel way of calibrating mid-range weather forecasts, it would be good to critically reflect on the results:

a.  Is the FWI a suitable predictor for fire events in all three regions? What are the challenges regarding wildfire hazard in the three regions? For example, you could address why the postprocessing works particularly well in the MED and summer months of WEU and why not for NEU? Further, you could reflect on how low FWI values, as present in NEU, affect your method.

b.  How does your method (NGR) compare to other post-processing methods? Select a two to three different studies, with a similar research question and set your results in a broader context. For example, you could discuss why you are correcting the FWI instead of the input variables of the FWI, why you chose the NGR method and not a bias correction method or machine learning based method (see Whan et al. 2021 https://www.sciencedirect.com/science/article/pii/S2212094721000086) and Worsnop et al. 2021 (https://journals.ametsoc.org/view/journals/wefo/36/6/WAF-D-21-0075.1.xml?alreadyAuthRedirecting)

c.  What can stakeholders take away from your study. You illustrated quite nicely in the introduction that post-processing helps to make accurate forecasts helping first responders. What do you wish this target group takes away from your results, e. g. will more firefighting resources be placed at locations with elevated FWI?

Thanks for this comment. We decided to incorporate the discussion in the conclusion, which reads now as follows:

*"We investigated whether non-homogeneous Gaussian regression (NGR) can be used to calibrate fire weather index (FWI) forecasts based on medium range ensemble weather forecasts by the European Centre for Medium-Range Weather Forecasts ensemble forecasts (ECMWF). To estimate the calibration coefficients of the NGR, we employ a truncated Gaussian distribution with cut-off at zero and use forecast and observation pairs of the last 30 days preceding the forecast. We used ECMWF high-resolution weather forecasts with the shortest possible lead time to calculate the FWI analysis and used these as substitute for observations. Although the FWI analysis seems to underestimate the FWI slightly, a good correlation is observed.*

*FWI forecasts using medium range ensemble weather forecasts perform generally quite well when compared to the analysis. However, calibration improves the forecasts especially at short lead times up to 84 hours. In the Mediterranean region and Central Europe an improvement of FWI forecast with respect to the FWI analysis is also apparent for longer lead times up to 8 to 10 days, respectively. This is likely caused by the generally higher values in those regions and is supported by the monthly averaged metrics in the appendix, which*

*show a stronger improvement caused by the calibration in the months with high FWI values. Hence, it can be concluded that the calibration performs better for higher FWI values, as indicated by the better performance during the summer months in West and Central Europe and in the European Mediterranean. However, the calibration method shows limitations for low FWI values, which could be observed in the NEU reference region and especially for longer lead times. Although it would be ideal for the method to perform effectively across the entire range of FWI values, it is generally more critical that it demonstrates a good performance at higher FWI values, where the potential fire danger is more significant. While long-range forecasts of potential fire danger are valuable, short-range forecasts, especially for the first 1-3 days, are usually more critical for firefighting resource management. Reliable and accurate forecasts, particularly when the fire risk is high and over the short-term time frame, are crucial for decision-makers and emergency responders to effectively coordinate resources. The improvement of FWI forecasts using the presented calibration method improves the ability to anticipate fire danger, ultimately supporting better response management and shows that a relatively simple method can provide good results compared to more complex approaches, e.g., bias correction (Cannon, 2018) or the correction of the input parameters instead of the FWI as in Worsnop et al. (2021). When correcting the individual input parameters, different models need to applied for the individual variables, which require careful verification and access to quality controlled observation over the whole stydy region. To further improve the presented calibration method for fire weather index forecasts, it could be tested if calibration of individual components of the FWI system e.g. FFMC, DMC and DC would improve overall skill of the forecast. Furthermore, more advanced models using additional predictors, e.g. elevation or land-use, could improve the calibration but were not tested here."*

The take-away message of our study is that NGR can improve FWI forecasts and therefore provide more accurate predictions of potential fire danger. In addition to being illustrated in the introduction, we again mention this now in the conclusion, also. We refrain from giving suggestions for possible actions based on the provided results, e.g. making suggestions for firefighting resources, as this is not the purpose of this article. However, supporting firefighting resources is the objective of the SAFERS EU project, which is mentioned in the introduction.

**Minor comments**

1. Line 6 & Line9: you use the terms post-processing and calibration interchangeable, please choose one term.

   Thanks for pointing this out. We decided to use the term calibration and changed this throughput the manuscript.

2. Lines 9 – 11: Be more specific about what you mean by short lead times (e. g. 84h?) and regions with elevated FWI (e. g. MEU)

   We included more specific information about the lead time and region as follows:

   "The calibration improves FWI forecast particularly at shorter lead times up to 84 hours and in regions with elevated FWI values, i.e., areas with a higher wildfire risk like Central and Mediterranean Europe. "

3. Lines 13: I would drop the word "recent" and replace "wildfire in Greece 2023" by "wildfire season of 2023".

   We changed the sentence according to your suggestion.

4. Line 15: Drop "Also" at beginning of sentence.

   We rephrased this sentence to avoid starting with "Also".

5. Line 16: Drop "But" or make this sentence sound more formal.

   We rephrased the whole sentence in order to sound more formal and to accommodate the added references (comment 6).

   "While the Mediterranean region continues to face the highest occurrence of wildfires, Central and Northern Europe have experienced an increase in extreme temperature events and heatwaves in recent years (Ibebuchi and Abu, 2023; Rousi et al., 2023; Ionita et al., 2017; Barriopedro et al., 2011). Extended warm and dry periods raise the fire danger and may cause wildfires in regions that were previously not considered wildfire hotspots (San-Miguel-Ayanz et al., 2021; De Rigo et al., 2017). Examples are the 2018 heatwave, which caused wildfires in Sweden (San-Miguel-Ayanz et al., 2019) and across the United Kingdom (Sibley, 2019) or the wildfire outbreaks in Germany and the Czech Republic during the summer of 2022 (Skacel et al., 2022)."

6. Line 17: Provide references for your statement.

   References for the increased extreme temperatures and heatwaves have been added to the statement (see comment 5).

7. Line 19: Provide references for your statement.

References are added (see comment 5).

8. Line 20: Here it would be great if more than one example could be provided, e. g. one for each subregion.

   We added two more examples, one of which is for the WCE subregion (see comment 5).

9. Lines 22 – 25: Switch the order of the sentence to stress more clearly that weather forecast is part of SAFERS or drop mentioning the project.

   We changed the order of the sentence to make it more clear that the weather forecasts are part of SAFERS.

10. Line 26: Add Di Giuseppe et al. 2016 (https://doi.org/10.1175/JAMC-D-15-0297.1) as a reference.

    The suggested reference has been added to this sentence.

11. Line 26: Watch out that your citation tool takes the names correctly. It is van Wagner and Di Giuseppe not Wagner and Giuseppe.

    All author names with surname prefixes have been checked and corrected.

12. End of line 27: Here I am missing a short explanation of what is the difference between deterministic and probabilistic weather forecast. A short explanation would be helpful to emphasize that the topic of the manuscript is postprocessing probabilistic forecasts.

    We added a short explanation of the difference between deterministic and probabilistic forecasts.

    "While deterministic forecasts provide a single forecast based on a given set of initial conditions, probabilistic ensemble forecasts offer a range of possible outcomes by using slightly perturbed initial conditions, giving a more comprehensive picture of potential weather conditions and providing an estimate of the forecast uncertainty."

13. Line 28: drop the word "may" or provide a clear statement whether post-processing is needed. Consider also my first comment on your interchangeable usage of post-processing and calibration. This confuses the reader.

We dropped the word "may" and changed post-processing to calibration.

14. Line 39: Chose a more scientific formulation than "is a relative simple calculation" for the FWI.

    We changed the formulation of this sentence to "*straightforward computation*".

15. Line 43: Add the depth of the moisture levels.

    The depth and type of fuel layer have been added. All changes to the FWI calculation section (2.1) can be seen in referee 1, comment 1.

16. Line 48: Rephrase the sentence starting with often, e. g. "The FWI can be classified"

    The sentence has been rephrased to:

    "The FWI is often classified into danger classes and values above 50 are considered extreme."

17. Lines 52 – 54: I would change the order of the sentences to make the statement at the beginning of the paragraph clearer, e. g. "we use climatological mean values …, to account for preceding conditions at the initialization".

    We follow you suggestion and changed the order of sentences.

18. Lines 59, 63 and 70: Please provide an approximation of the grid resolution in km in brackets?

    An approximation of the grid resolution in km has been added in those locations.

19. Line 60: it should be "derived from the TIGGE archive".

    The article has been added.

20. Line 61 and following: Please add dataset after TIGGE, i. e., "The TIGGE dataset …".

    We completely rewrote the forecast and observation data section (2.2) and don't refer to the TIGGE dataset in those sentences anymore.

21. Line 62: Please add "the" to ECMWF API.

We added the article "the".

22. Line 62: Please rephrase sentence to "the temporal resolution of the TIGGE dataset …".

We rephrased this sentence to:

*"The temporal resolution of the ensemble forecast data used in this paper is 6h for all lead times and the spatial resolution is 0.5° (~50 km)"*

23. Lines 65 – 78: Please rephrase this paragraph. Keep the statement that the FWI has multiple input variables. Explain for which later steps you use which dataset to calculate the FWI. Mentioning the station data here is confusing.

We rephrased the whole section 2.2 (see major comment 4) and hope it now clear which dataset is used for which step. The station data is now mentioned in the second part of this section after introducing all forecast datasets.

24. Line 71: I am not sure which dataset you are meaning by "those", please clarify.

Section 2.2 has been completely rewritten and it should now be clear which dataset is meant.

25. Line 75: How many of the 682 stations are in Finland and how many are outside of Finland. Can you provide values.

Because the correlation coefficient of FWI analysis and FWI observation is basically identical for Finland and the stations outside Finland and no additional conclusions can be drawn by separating Finland from the rest of the stations, we decided to merge the dataset. It is therefore no longer necessary to mention the number of stations in Finland separately.

26. Fig 1 / Line 74 (first mentioned):
    o Add letters for subfigures.
    o Fig 1a
        ▪ Use a different projection, e. g. Lambert Conformal Conic, as the northern latitudes are strongly distorted.
        ▪ It would be very nice to have the stations shown in Fig 1c on the map of Fig 1a as well.
        ▪ Place the region legend (i.e., NEU, WCE, EUMED) inside Fig 1a.

- Place the legend of the countries (i.e., Finland and others) at a position, where it is clear the legend belongs to both Figures (i. e. Fig 1a and Fig1b)

We implemented all suggested changes in Fig. 1a.

[Figure]

*Fig1*

- o Fig 1b:
    - Provide a legend for the regression line. Is this the line for all stations, or for only "other" stations (outside of Finland) or for only Finland stations?
    - Can you provide a line for Finland as well?

There were 2 regression lines in Fig 1b, one for Finland and one for the other stations. Those regression lines were basically identical and therefore not very well visible. Figure 1b has completely been changed according to suggestions of referee 1 (major comment) and we now show a separate Figure (Fig.2) with histograms of the FWI values and correlation coefficients.

[Figure]

*Fig.2*

- o Fig 1c:

- Please show a 3rd station for WCE.
- Add the location of the stations in Fig 1a.
- Why are there missing values in the Greece station in the winter of 2022 and 2023? You previously stated that all your selected stations have a sufficient data coverage. Please clarify that your consecutive 200 days refer to the summer half (?) in the Figure caption.

We added a third station (Meiningen, Germany) as an example for WCE and marked its location on Fig. 1a. Accessing data of stations outside Finland with year-round continuous data proved challenging. To expand our analysis to stations outside of Finland, we included stations with some missing data, as long as they had at least 200 continuous days of data. This period isn't restricted to summer months. We chose this 200-day threshold to avoid too frequent reinitialization of the FWI calculation, which could affect the comparison of observation and analysis.

27. Line 77: Provide a reference to Fig 1b as your statements originate from the figure.

Fig 1b has been updated and mentioned scatterplot is no longer shown in the updated manuscript.

28. Line 77 and Fig 1b: Is your, I assume linearly derived correlation, mainly driven by the large number of low FWI values? Fig 1c shows quite apparent that for high FWI values the underestimation is much stronger pronounced than for low FWI values. This would be also a good point to be discussed in the discussion.

The correlation is indeed mainly driven by the large amount of low FWI values, which is one reason we decided not to present the scatterplot and linear correlation in the revised manuscript. Instead, we show the distribution of FWI values calculated from observation and analysis using histograms (Fig. 2a,b) and the distribution of the linear correlation coefficient in Fig.2c. Furthermore, we point out that the especially for high FWI values the analysis is underestimating the FWI with regard to the observation when referring to Fig.1 b,c,d.

29. Line 79: Provide a reference to Fig 1c.

Fig. 1c has been updated to Fig.1 b,c,d. References to the respective figure are added in the revised manuscript.

30. Line 80: Please clarify which datasets you use for longer forecasts.

We use the medium-range ECMWF ensemble forecast and clarified this in the revised manuscript.

31. Line 83: Please clarify which dataset(s) you mean by the FWI forecasts.

We added "*medium-range*" to the FWI forecasts to clarify which forecasts we mean. With the changes made in Section 2.2, it should now be clearer which dataset we refer to.

32. Line 86: Please specify what you are calibrating, e. g. the parameters of the NGR or the whole post-processing pipeline.

We added that we are calibrating the FWI forecasts to this sentence.

33. Line 86: It should be: "the FWI".

We added the article.

34. Line 91: Please clarify from which dataset the 51 ensemble members come, e. g. by adding (ENS) in brackets. You describe the statistical part very clearly, but it is not clear to which datasets you are applying the formulas.

The ECMWF ENS forecast is no longer mentioned in the revised manuscript. With the changes made to Sect.2.3.1, it should be evident that the calibration is done for the FWI calculated from the ECMWF forecasts (derived from TIGGE database).

35. Line 96: Please specify what the training area is.

In this study the AR6 WGI reference regions are used as training areas. We agree that the training areas need to be introduced already when explaining the NGR method and therefore moved the introduction of the AR6 WGI reference regions from the results section to Sect. 2.3.1.

*"The training data is defined using a specific training area and a 30-day rolling window preceding the forecast as training period. As training areas, we use here climatic reference regions, defined by the 6th IPCC Assessment Report (AR6-WGI (Iturbide et al., 2020)), which divide the European domain into Northern Europe (NEU), West and Central Europe (WCE), and the Mediterranean (MED). In this study, however, we use only the European part north of the Mediterranean Sea, referred to hereafter as the European Mediterranean (EUMED). The reference regions are*

*marked grey in Fig.1(a). The calibration can also be performed over smaller geographical areas, e.g., individual countries."*

36. Line 97: Why do you switch terminology from NGR to EMOS, I understand that this is the approach, but it would be good to decide for one name.

   We removed the reference to EMOS to avoid confusion.

37. Lines 98 -100: Can you provide results, e. g. a table or a plot, for these findings. This could be part of your supplementary material.

   We added an additional figure in the Supplementary material Fig. S1. This figure shows the average CRPS for a test time period (June-July 2021) using different training areas.

38. Line 100: Here it becomes apparent, that it is not clear what the training area and hence smaller geographical training areas should be. Please clarify and provide results in the supplementary material.

   In the revised manuscript, we now introduce the concept of training areas prior to this statement (see comment 35) and clarify that smaller geographical regions can also be used for calibration purposes. We hope this revision makes the meaning of training areas clear.

39. Line 114: Please add a note that you call RMSE later spread and skill metric (i. e. line 122 and line 144).

   This has been clarified in the text.

40. Line 142: please clarify that you compare fire season length of Northern Europe to Southern Europe.

   We added the note that we compare Northern and Southern Europe here.

   *"The main fire season in Europe is typically from May until October but varies strongly in length and intensity across regions, e.g. the fire season starts later and is shorter in Northern Europe compared to Southern Europe (San-Miguel-Ayanz et al., 2012)."*

41. Line 145: the grid points "within" rather than "of" the three study areas.

*We changed the wording accordingly.*

42. Line 146: I can't follow how you derived the RSME of the climatology. Can you describe this briefly.

*We added a brief explanation how the climatology RMSE is calculated as follows:*

*"The RMSE of the climatology, shown by the solid blue line, is calculated similarly to the forecast RMSE. However, it uses the climatology derived from 40 years of ERA5 reanalysis data (see Sec.2.2) for each day of the year instead of using the forecast."*

43. Line 151, Line 163: clarify that your calibration is done as a post-processing.

*This is now clarified in the first sentence of the Results section:*

*"In this section, we present results of post-processing FWI forecasts of the years 2021 to 2023 by applying the introduced calibration method."*

44. Line 152: Specify what you mean by short lead times, e. g. 132h?

*In this specific case, we decided to drop the part of the sentence mentioning short lead times. Short lead times refer only to the NEU region (~132h) while the RMSE of the calibrated forecast is still smaller than the RMSE of the raw forecasts up to 228h for WCE and 300h for EUMED. These lead times are mentioned in the subsequent sentences.*

45. Line 154: Provide lead time in hours in brackets after "7 days", i. e. 7 days (168 h).

*We included the lead times in hours in brackets after every following mention of the lead time in days. It has to be noted that the lead time in hours doesn't exactly match the time in days because the FWI forecast is done for 12 UTC using the forecast initiated at 00 UTC. The first day's forecast has therefore a lead time of 36 hours and not 24 hours.*

46. Figure 2:
   - Add labels (letters a, b, c) to subplots.
   - Place legend outside of NEU to make it clear it belongs to all three subplots.
   - Adjust the y-axis label (Spread/ RMSE) to your figure caption, which is currently "spread and skill". I would expect them to be the same, e. g. spread and RSME in the figure caption or spread and skill in the y-axis label.

All suggestions have been applied to Fig.2 (now Fig.3). We adjusted it to be Spread/RMSE on caption and y-axis label.

[Figure]

*Fig.3*

47. Line 157: Rather the three subregions than the respective area.

The sentence has been rephrased accordingly.

48. Line 158: Rephrase "too low" to something like "lower than observations".

We rephrased the sentence to:

*"Uncalibrated forecasts have a negative bias for all lead times, indicating that the forecasted FWI values are consistently lower than the observed values."*

49. Line 160: "The improvement" instead of "This improvement".
We changed the article accordingly.

50. Line 164: Provide numbers for what you think is slightly positive.

We provide numbers of the positive mean error now.

"In Northern and Central Europe, the bias is slightly positive after calibration especially for longer lead times, ranging from less than 0.1 to 0.4."

51. Line 167: specify short lead times.

We specified here that lead times up to 84 hours are considered short lead times.

52. Line 166 – 169: This finding would be a good point to discuss in the discussion. For example, I would be interested what these findings imply for the application of your suggested post-processing technique.

    This has been added to the discussion in "4. Discussion and conclusion", see major comment 7.

53. Figure 3: please add letters to subregions.

    The abbreviation of the subregions has been added to the caption.

54. Figure 4: Please add letters to subregions.

    The abbreviation of the subregions has been added to the caption.

55. Figure 4 caption: drop "the grid of".

    "the grid of" has been dropped.

56. Lines 171 – 172: Can you discuss this in your discussion section? Does your approach perform well for higher FWI values, suggested by the better performance in WEU in July and August and MED? Does your approach need to perform well on or low no-fire danger days?

    This has been added to the discussion in "4. Discussion and conclusion", see major comment 7.

57. Figure 5:
    o Add labels to the subplots (i. e., a, b, c, d)
    o Plot the land-sea boundary to improve the visualization.
    o Drop the large white space in the south and west of the plot in such a manner that the plot is filled with results.

    All suggestions to improve the visualization of Figure 5 (now Figure 6) have been implemented, additionally we changed the map projection to be in accordance with the map in Figure 1.

[Figure]

*Fig.6*

58. Line 176: Here you mention the first time that you calibrate the coefficients of the NGR, this is not clear in your previous description of the post-processing method. Please clarify this in the method section. In Line 176 add "of the NGR" after "to estimate the calibration coefficients".

    We added "of the NGR" to the sentence. The method section has been improved and is hopefully clearer now.

59. Line 177 - 179: This sentence belongs to the discussion section and not the conclusion session. Also, I suggest adding more meaning to this sentence, e. g. what are the implications of sparsely available data?

    We removed this sentence from the conclusion.

60. Line 180: Drop "Thus".

    "Thus" has been dropped.

61. Line 180: which dataset do you mean by high-resolution weather forecast with short lead time? Is this the third dataset you introduced in the data section?

Yes, the high-resolution forecasts are the third dataset that is introduced. We restructured the Forecast and observation data section (now Sec. 2.2.) and hope it is now clear what we mean with high-resolution data throughout the manuscript.

62. Line 181: Make clear that you mean the dataset "analysis" and not the analysis of the FWI.

We added a reference to the previous sentence to make clear that we mean the FWI calculated by the forecast when referring to analysis.

63. Line 189: At the end of your conclusion, I would expect a last sentence coming back to your initial statement that this is important for fire resource management and the SAFERS project. Please add such a sentence.

We added a sentence to the conclusion that highlights the significance of our findings for fire resource management as follows:

"Reliable and accurate forecasts, particularly when the fire risk is high and over the short-term time frame, are crucial for decision-makers and emergency responders to effectively coordinate resources. The improvement of FWI forecasts using the presented calibration method improves the ability to anticipate fire danger, ultimately supporting better response management."

64. Line 209: Add "Di" to "Di Giuseppe".

We fixed this mistake in the reference.

65. Line 249: Add "Van" to "Van Wagner".

We fixed this mistake in the reference.

---

## Referee Report (RR1)

The authors improved the manuscript and addressed the reviewers' remarks in most parts of the manuscript. However, the manuscript still lacks consistency in terminology and appropriate order of arguments/ statements to have a good reading flow. A critical discussion of the study's results is still missing. The authors point out flaws in their results quite clearly, but do not set them in context with other studies using similar methods, datasets, or parameters.

These issues are of minor magnitude but should be addressed. Therefore, I am happy to read the revised manuscript once again after the following points have been addressed:

**General minor comments**

1. Terminology
   a. Be consistent in how you refer to figures in your text (see comment 22): You use Figure 2a (line 107), while in line 99 you use Fig. 1 and 2. Also make sure the blank space is consistent between figure and figure number as well.
   b. Name your metrics consistently, e. g. RSME and skill are used interchangeably. I pointed it out in comments 35,39 and 41 but there might be locations I oversaw. Please check this carefully before resubmission.
2. Definitions:
   a. You use relative statements throughout your paper. Make clear what are low / high FWI values or what is a good correlation. Add numbers to these relative terms and state why you use these numbers as thresholds to define a FWI to be high or low, or a correlation to be good (see comments 20 and 24)
   b. Same for short- and long-term forecasts. Define these terms in your text, e. g. short term (up to 3 days) and long term (more than 3 days).
3. General comment on verification and Fig. 2b:
   You state several times that your approach does not work well for small/ low FWI values. What are the thresholds for you to consider FWI to be low? From your study I read that this is below 1, but from the EFFIS danger classes ([https://forest-fire.emergency.copernicus.eu/about-effis/technical-background/fire-danger-forecast](https://forest-fire.emergency.copernicus.eu/about-effis/technical-background/fire-danger-forecast)) low FWI values are considered to be below 11.2. Your underestimation of the FWI in the analysis mainly affects this relevant scale (10 -100) (see Fig 2b).
   - Can you mention this in the description of the figure (i.e. line 110)
   - Discuss and state stronger that your calibration does not adress this discrepancy since you calibrate on the (orange) analysis and not on the (grey) observations density?
   - What are the impactions of your decision to use the analysis as observations in the calibration process. Don't you systematically underestimate large/ relevant FWI (FWI > 10) values by calibrating to the analysis over observations? Please add your elaboration on this to the discussion and conclusion.

**Detailed minor comments**

1. Line 2: "makes accurate wildfire risk estimation crucial" add a target group, e. g. decision makers, emergency responders, etc.

2. Line 6: You already introduced the abbreviation FWI for the Canadian Fire Weather Index. I suggest to not spell it out here and just use FWI.

3. Add a strong last sentence to your abstract, in which you point out what the target group and the impacts of your study should be.

4. Lines 13 - 21: You added good references for your description of the more northern parts in the second part of your paragraph (line 16ff.). Could you underline your statements in the first part as well?

5. Line 27: In your abstract you say you look into forecasts up to 15 days which is 2 weeks, while here you state you are interested into "forecasts ranging from a few days to several weeks". Do you refer to your study or the outline of the SAFERS project here and what are several weeks?

6. Line 28: Rephrase the sentence and add what you use the FWI for, e. g. "in this paper, we use the FWI … for deriving fire risk from weather forecasts".

7. Line 28ff: Here you could improve the reading flow by two things:
   o You should shortly mention the parameters of the FWI, i.e. naming the four weather parameters in it (temperature, precipitation, wind speed and rel. humidity).
   o Add a transition to weather forecasts by adding a statement that these parameters are available in deterministic and probabilistic forecasts.
   o Then you can continue with explaining the pros and cons of deterministic vs. probabilistic forecasts.

8. Line 35ff: Thank you for adding this paragraph about other methods and adding your purpose of the study. This section is much clearer now.

9. Line 49: Add a reference after various regions, e. g. Di Giuseppe et al. (2016)

10. Line 51: Please rephrase "relatively straightforward computation" to a more scientific language. I disagree that the many empirical formulations of the FWI and its subindices are straightforward to develop, though they are straightforward to apply once a source code is available.

11. Line 55: Add ", i.e. Drought Code (DC), Drought Moisture Code (DMC), and Fine Fuel Moisture Code (FFMC)".

12. Line 65: Add ", i.e. Initial Spread Index (ISI)."

13. Line 70: Add reference to EFFIS fire danger classes.

14. Line 70-71: I like that you reflect on the fire danger level thresholds here, but you do not show results for FWI values later in your results section and you do not group your FWI values into these classes. Therefore, you can remove these two sentences about the fire danger levels.

15. Line 74: Drop "with modifications to utilize gridded input data" or provide the code in a GitHub repository, because I am curious to see/ use it now.

16. Line 86: You missed a word ("mean"?) after climate.

17. Line 87: Rephrase to "using a centered 15-day rolling mean on each day".

18. Line 89: Here, I do not understand where the ECMWF high-resolution forecast comes from. Is this part of the TIGGE archive or from a different archive? Please clarify.

19. Line 93ff: This sentence is confusing. Aren't you showing weather station data in Fig. 1a)? Please restructure this in accordance with the beginning of your new paragraph and make this a uniform/ clear statement.

20. Line 103: Please put your correlation coefficient (r=0.72) in brackets here. The definition of a good correlation is varying in different science domains, and I would rather read that you found a correlation coefficient of 0.72 and why you think this is sufficient for your analysis.

21. Line 107: Missing word after Figure 2a.

22. Line 107 and wherever you refer to a figure, here you use Figure 2a, while in line 99 you use Fig. 1 and 2. Please make this uniform across your manuscript.
23. Line 112: Do you mean analysis by "forecast-derived"? Please stick to one term across your manuscript.
24. Line 115: Why is it here only a "fairly good" correlation while it was a "good correlation" in Line103?
25. Line 119: Make the statements about the markers more specific, e. g. "the bright red colored markers".
26. Fig. 1b-c legend + Fig 2 legend: shorten "calc. from observation" to just "observations".
27. Fig. 1 caption:
    o "See Fig. 2" to "see Fig. 2c".
    o Add "(b – d)" after example stations.
    o Move "(b)" in front of "in NEU".
28. Fig 2c: Can you add the rmean as a vertical line to the plot?
29. Line 143 – 146. These two sentences need to be restructured. Mention the length of your training periods in the beginning, e. g. "training periods between 15 to 40 days were tested", then add your benefits and disadvantages of different training lengths.
30. Line 147 – 148: What training period are you using in your analysis? You say you found 30 days appropriate for small geographical areas and then that you adopt a regional approach, which is in my understanding a large geographical area. Please clarify which training period you stick with here explicitly.
31. Line 158: Please replace several with the verification metrics you are using.
32. Line 160: Drop introductory sentence.
33. Line 164/ 165: I don't understand how you derive the ensemble spread. Is the ensemble spread the standard deviation (square root of the ensemble variance) of the ensemble? What is the average ensemble variance? The mean of the variance of all ensemble members? Please rewrite this.
34. Lines 166/ 167: Please be clear in your terminology. Is skill in line 167 the same as RMSE in line 166? Please be consistent with these terms throughout your manuscript and figures, i.e. Fig. 3.
35. Line 185: Replace "introduced calibration method" with "the NGR method".
36. Line 186: Name the regions again, i.e. NEU, WCE and EUMED.
37. Line 190: What is a considerably high FWI? Please find a reference using this fire seasons to refer to your Fig 1b-d.
38. Line 197-199: Here it becomes apparent that you should be consistent with naming it either skill or RMSE.
39. Line 200: Move the statement that your method improves the forecast to the beginning of the sentence to make your statement stronger.
40. Line 202/ 203: Consistent naming of RMSE and skill becomes apparent again. From the reading flow it might make more sense to stick to RMSE when you talk about the measured RMSE (line 202: "skill of raw and calibrated") and use skill when you talk about the performance of your calibration method (line 203: "the forecast lacks skill").
41. Line 205: What do you mean by large uncertainty? If you mean that the signal-to-noise ratio in NEU is higher than in other regions, because the FWI values are lower and more unlikely to exceed the variability range, then bring this statement more to the point. You could underline this statement by whatever you mean with "(not shown here)".
42. Line 206: What are you not showing here and where do you show the relative uncertainty in different regions?! Drop this statement, refer to supplementary material or refer to a Figure.
43. Line 214: Just add in brackets the reference to the figures in your supplementary material.

44. Line 222: What is your reference in Fig. 5 for stating that the skill worsens in NEU but not in WCE? In Fig 5. the median in WCE goes below 0 at lead time between 180 and 220 and therefore, I think that the skill worsens here as well. In EUMED the skill is not worsening if you refer to the median of the boxplots.

45. Line 223/ 224: I don't understand the context of these two statements: Is this worsening of skill caused by small FWI values or by longer lead times? Didn't you say earlier you excluded FWI values below 1? Can you refer to a figure why you think low FWI values are the cause of the worsening? Throughout your manuscript you bring up this argument frequently, but you never show how high/ low the FWI values in the different regions are, and why you come to this conclusion.

46. Line 225: Merge this with the paragraph above. Shorten this by dropping the first sentence and adding the second sentence to the paragraph above by referring to the figure in brackets: "With increasing lead time the skill worsens especially in mountainous areas in Scandinavia and the Alps, which are also the regions with generally low values throughout the fire season (see. Figure 6)."

47. Figure 6 comment: You claim that your method does not perform well over mountain regions. Which is true for lead times of 228h and 324h. But at lead time 132h, I see that the atlantic influenced regions all do not perform very good. Has this something to do with the ocean/land interface? Could you investigate this and add a statement about this to your manuscript?

48. Line 234: please shorty reintroduce the term FWI analysis, e. g. FWI derived from the ensemble forecast (analysis). For people skimming your paper it might appear as you analyze the FWI, which you don't.

49. Lines 236 -239: Please be consistent when using lead time units. It is hard to transfer from 84 hours to 8 to 10 days. You can state the lead time in hours and add the days in brackets after the hours as you stated in your response to the reviewers.

50. Line 240: this statement is weak and a reference to appendix (e.g. "see Fig S5-7") is missing. From these figures you can only read that the mean error is smaller in the summer months, but not how high or low the FWI is in the specific subregions. Therefore, you cannot state here that your findings are likely caused by low FWI values.

51. Lines 240: It is not clear what you consider high and low FWI values, please add values or define low or high FWI somewhere, e. g. "high FWI values (FWI > 1)".

52. Line 244: Rephrase "Although it would be ideal" to more scientific language.

53. Line 246: Define what you mean by "long-range" and "short-range" forecasts.

54. Line 251: Move everything after "compared to more complex approaches" to the beginning of the discussion and conclusion chapter, e. g. to line 233 before you start with the sentence "We used...", to end your paper with this very good and strong statement: *The improvement of FWI forecasts using the presented calibration method improves the ability to anticipate fire danger, ultimately supporting better response management and shows that a relatively simple method can provide good results compared to more complex approaches*. In line 233 you can discuss that NGR is a good method in comparison to other methods (e.g. bias correction) for your research purpose.

---

## Author Response (AR3)

**Author's response**

Thank you for reading the manuscript and providing valuable suggestions to improve the paper. Our responses to the reviewer comments are shown below in blue, and changes to the manuscript are indicated in italics. Additionally, all modifications are marked in the revised manuscript.

**Reply to the comments of reviewer 1**

**Major comments**

45-47 ". In the second step, FFMC, DMC and DC are used to model the rate of fire spread (ISI) and the potential fuel available for surface fuel consumption (BUI)." The wind speed is missing (ISI = FFMC + WS10), and it would correlate to the POTENTIAL rate of spread (I mean, it is a variable involving the combustion of surface fuel, like dry leaves and such + the wind). Also, the BUI = DMC + DC, so it is not surface fuel consumption, but it involves the potential for a surface fire to burn the deeper fuel (Build Up) and become a much more persistent fire.

Thank you for your comment. The suggestions have been included in the revised manuscript. The description of the FWI calculation reads now as follows:

*"The FWI is calculated in two steps. First, the 2-meter temperature, 2-meter relative humidity, 10-meter wind speed, and 24h accumulated precipitation at local noon are used to calculate the moisture content of three separate fuel layers. These fuel layers are characterized by different depths and fuel consistencies, which result in varying water capacities and drying speeds. The Fine Fuel Moisture Code (FFMC) represents the moisture content of litter and other fine cured fuels at a nominal depth of 1.2 cm; the Duff Moisture Code (DMC) indicates the moisture content of loosely compacted layers (nominal depth ~7 cm); and the Drought Code (DC) denotes the moisture content of deep, compacted layers in a depth of around 18 cm (Van Wagner, 1987). DMC and DC respond slower to weather variations compared to the fast-drying fuel represented by the FFMC. Consequently, the effective day length, which determines the amount of drying that can occur during a given day, must be considered and monthly day length adjustment factors for DMC and DC are applied based on latitude (Lawson and Armitage, 2008). The fuel moisture codes are dependent on previous weather conditions; therefore, the preceding day's noon values for FFMC, DMC and DC are necessary for their calculations. In the second step, FFMC and the 10-meter wind speed are used to model the potential rate of fire spread (ISI). DMC and DC are used to calculate the Buildup Index (BUI), a numeric rating of the total amount of fuel available for combustion which comprises the potential of a surface fire to burn deeper fuel layers (build up) and thus evolve into more persistent fires. These fire behaviour indices are then used to calculate the FWI. The FWI values are always non-negative, with low numbers indicating low fire weather danger and high values indicating high fire weather danger. The FWI is often classified into danger classes and values above 50 are considered extreme. However, those levels can vary depending on local conditions (e.g., vegetation types).*

*Consequently, what is considered a low or extreme FWI in one region may not be equivalent in another. A more comprehensive description of the FWI system can be found in Van Wagner (1987) and Lawson and Armitage (2008). For this study, we implemented the calculation of the Canadian Forest Fire Weather Index (FWI) using Python programming language, following the source code provided by Wang et al. (2015), with modifications to utilize gridded input data."*

76-78 "Figure 1b shows the scatter plot of analysis and observations for all stations and every time step. While the FWI derived from the forecasted weather parameters seems to generally underestimate the FWI values compared to the values derived from the observations (slope ∼ 0.63)" AND Figure 1b: Many doubts arise from this scatterplot:

- The plot itself shows too many points to use a scatter. A density plot NEEDS to be used in this case, or two if you want to show separately the data from Finland.

Thank you for your comment. We agree that the scatterplot may not have been the most suitable choice for representing our data and conveying our message. We decided to change the representation to histograms showing the distribution of the FWI calculated from high-resolution forecast (analysis) and from observation (Fig.2 in the revised manuscript). Additionally, we split the Figure into 2 parts to avoid overcrowding the figure. The new Fig.1 contains the map of observation stations (a), with colour representing the linear correlation coefficient, and the time series of three example stations (b,c,d). The new Figure 2 contains the histograms of the frequency distribution of the FWIs (a,b) and the linear correlation coefficient (c).

- The regression which leads to the 0.63 slope seems to be off by looking at the scatterplot, which might be due to the lack of information about the point density. What causes the slope to be 0.63 (and not closer to 1, as the scatterplot would suggest)? Also, please specify the method used for the regression (I assume linear regression).

The regression line in the original Figure 1 appeared off due to the high concentration of values near zero. This becomes now evident considering the distribution shown in histogram Fig.2a. We also clarified in the text that we used linear regression to obtain the coefficient.

- I am aware that it is common practice to use the ECMWF analysis at minimal lead time in place of observations, but once you have found an important underestimate like you did, why did you dismiss it so fast? It seems like a very important matter that can have a huge impact on the paper's reliability. Please explain in depth why you can ignore this bias or what you did to correct it.

We agree that a correction of the analysis based on the observations would be beneficial. We found that the strongest discrepancies between observation and analysis occur in mountainous areas, e.g., Austria or Norway. This is likely caused by difficulties in capturing the small-scale weather phenomena occurring in the complex terrain with relatively coarse resolution of the forecast model. However, in those regions the fire danger is generally rather small because of high precipitation and low temperatures. To further investigate this bias, additional station-specific characteristics such as elevation or land use would be

necessary. In this article, we want to present an easy, straightforward method to calibrate the FWI and we believe that the correlation of analysis and observation is already quite good and justifies the use of the analysis as substitute for observations. However, the use of additional station-specific parameters could be topic of future research.

The section that previously related to the scatterplot has been changed to:

*"To determine if the analysis is suitable to be used as observation substitute, we check their agreement with actual observation-based values, which is shown in Fig. observation and 2. We use observations available from the Finnish Meteorological Institute's observation database for the years 2021–2023. Figure 1a shows the stations, for which it is possible to calculate the FWI for more than 200 consecutive days in addition to the reference areas which are introduced in Chapter 3. In total 682 stations can be used. The time series of three example stations are shown in Fig.1a,b,c. The FWI analysis is shown in orange while the FWI calculated from observations is shown by the black dashed line. Overall, there is good correlation between the forecasted and observed FWI values. However, especially for high FWI values, the FWI derived from forecasted weather parameters tends to underestimate the values compared to those derived from observations. This is particularly evident during the summer months of 2022 and 2023 in Meiningen, Germany (Fig.1c) and Chrysopouli, Greece (Fig.1d).*

*Figure 2a overlapping histograms of the density distribution of the FWI analysis (orange) and FWI calculated from observations (grey) for all stations shown in Fig.1a and every time step. The distributions show a high frequency of low values (FWI< 1), and when plotted on a logarithmic x-scale, a bimodal structure becomes evident with a separation at an FWI value of 1. This bi-modality results from a necessary restriction imposed on the FWI calculation (Eq. 30a,b Wang et al. (2015) or Eq.40 in Van Wagner (1987)). Figure 2b again displays the density distribution of FWI values, but focuses on FWI values greater than 1, as these are generally more relevant for assessing wildfire danger. For higher FWI values, the forecast-derived FWI underestimates those calculated from observations, while for lower FWI values, it tends to overestimate. One contributing factor is the different spatial resolution of the data sources. The analysis uses gridded data with a resolution of 0.1° (approximately 9 km in Central Europe), whereas weather observations are local measurements. However, in general there is a fairly good positive correlation of FWI analysis and FWI calculated from observations. Figure 2c shows the histogram of the linear correlation coefficient of stations that observe FWI values greater than 1, the mean correlation coefficient is 0.72. Low correlation coefficients are mainly caused by differences between analysis and observation in mountainous areas, such as Austria, Switzerland, Romania, and Norway, as indicated by the colored markers in the station map (Fig. 1a). This is due to the difficulty in capturing the complex terrain and its small-scale weather phenomena with relatively coarse resolution (~9km) of the forecast model. Because there is a fairly good correlation of observation and analysis, we will use the forecasted FWI with short lead time (analysis) derived from ECMWF high-resolution forecasts as observation to compare with the FWI forecasts derived from ECMWF medium-range ensemble forecasts."*

[Figure]

Fig.1

[Figure]

Fig.2

78-79 "a correlation is apparent. This good correlation can also be seen in the time series examples for a station in Finland and Greece": Please provide us with the necessary quantitative information (e.g., correlation coefficients for all the stations) to support this claim, especially coming right after the previous comment. Two sample stations (Figure 1c) are not enough to validate a claim on over 600 others; by this I do not mean that Figure 1c must be removed.

We agree that only showing two examples is not enough to show that there is a good correlation. In the revised manuscript we included an additional figure (Fig.2c) which shows the distribution of the correlation coefficient for all stations. The mean correlation coefficient is 0.72, which indicates a fairly good positive correlation. Additionally, we added a third station in Fig.1, providing an example for each subregion. Changes to the manuscript can be seen in the reply to the previous comment.

90-97 "Furthermore, data from all grid points in the training area is used to estimate a single set of coefficients for the given day (regional EMOS)": some additional explanation is then needed, how do you go from the $\mu_{kl}$ to the estimate used (I guess $\mu_l$ ?)

We improved this section (now 2.3.1) to make the method better understandable and to clarify what training data is used. Specifically, we changed this sentence to:

*"We adopt here a regional approach, pooling training data from all grid points within the training area to derive a single set of calibration coefficients ($a_l$ - $d_l$) specific to each lead time for the given forecast."*

We hope this clarifies that the calibration coefficients are estimated by the calibration method and then used to calibrate the ensemble mean of the given forecast.

Figure 2:

- It can be made clearer if the legend was more explicit (dotted line with triangles: spread (raw) / solid line with dots: RMSE (calibrated) / etc. )

We made the legend more explicit as suggested.

- The legend, which is relative to all the three graphs must be outside the first graph. Consider also putting everything in a single column.

We placed the legend outside as suggested.

The previous Figure 2 is now Figure 3 in the revised manuscript:

[Figure]

*Fig.3*

**152-153 "In Northern Europe, the RMSE of the calibrated forecast is slightly above the RMSE of the raw forecast after 7 days of forecast," : please provide at least a hypothesis as to why this happens. The sentence on the subsequent lines "The regional differences could be explained with the generally higher FWI values in the more southern, fire prone regions compared to Northern Europe where FWI values are often very small" addresses the regional differences, but the difference from uncalibrated and calibrated NEU RMSE is not addressed.**

We added a hypothesis trying to explain the slightly decreasing RMSE of the calibrated forecasts in NEU as follows:

*"In Northern Europe, the RMSE of the calibrated forecast is slightly above the RMSE of the raw forecast after 7 days (180 hours) of forecast, whereas the skill of raw and calibrated forecast in Central and Mediterranean Europe is similar for forecasts longer than 9 days (228 hours) and 12 days (300 hours), respectively. The FWI forecast in Northern Europe lacks skill for lead times longer than 7 days and calibration based on 30 day rolling window fails to improve and even worsens the forecast. This finding could be explained by the rather small FWI values and large relative uncertainty in NEU, unlike FWI values in WCE and EUMED (not shown here). The applied calibration appears to be more effective for higher FWI values and shows limitations for smaller FWI values close to 0. This also explains the regional differences and the better calibration results in the more southern, fire prone regions with generally higher FWI values compared to Northern Europe, where FWI values are often very small."*

**Figure 3:**

- The legend, which is relative to all the three graphs must be outside the first graph. Consider also putting everything in a single column.

The legend has been placed below the figure to make it clear that it relates to all three subplots. Figure 3 is now Fig.4:

[Figure]

*Fig.4*

**Minor comments (typos and formalities)**

13: "prevalent": word choice

We changed the wording to "*frequent*".

16: "But not": cannot start a sentence with "but not"

We rephrased this sentence as follows:

*"While the Mediterranean region continues to face the highest occurrence of wildfires, Central and Northern Europe have experienced an increase in extreme temperature events and heatwaves in recent years (Ibebuchi and Abu, 2023; Rousi et al., 2023; Ionita et al., 2017; Barriopedro et al., 2011)."*

Furthermore, we added references to support the statement.

18: unnecessary comma after "periods"

The unnecessary comma has been removed.

19 "heatwave 2018": either heatwave of 2018 or 2018 heatwave

We changed it to *"the 2018 heatwave"*.

24 missing (Oxford) comma after "during"

The comma has been added.

24-25: "Accurate and reliable weather forecasts ranging from a couple of days to multiple weeks to identify high wildfire risk areas is an important part of SAFERS": Accurate and reliable weather forecasts, ranging from a couple of days to multiple weeks, are an important part of SAFERS for identifying high wildfire risk areas. (Or equivalent paraphrasis)

We rephrased this sentence and changed the order of the sentence to make it more clear that the weather forecasts are part of SAFERS.

*"An important component of SAFERS for identifying high wildfire risk areas are accurate and reliable weather forecasts, ranging from a few days to several weeks."*

25 "Here,": "in this paper,"

Wording has been changed to *"in this paper"*

26 "short FWI": (FWI)

This has been changed.

26: "Wagner, 1987": Author's last name is Van Wagner, throughout the paper

Thank you for pointing this out. This was a mistake in the used Bibtex reference and has been fixed.

29: "One widely used": A widely used

A paragraph mentioning other calibration methods has been added to the introduction following the suggestion of reviewer 2 (comment 2). The wording has therefore changed:

*"Various methods have been developed for calibrating probabilistic ensemble forecasts. Commonly used calibration methods include Bayesian model averaging (Raftery et al., 2005), non-homogeneous Gaussian regression (Gneiting et al., 2005), logistic regression (Hamill et al., 2004) and non-parametric ensemble post-processing methods such as rank histogram techniques (Hamill and Colucci, 1997), quantile regression (Bremnes, 2004) and ensemble dressing approaches (Roulston and Smith, 2003). Non-homogeneous Gaussian regression (NGR) is one of the most commonly used calibration methods and adjusts both ensemble mean and spread, while still be efficient and easy to implement. It has been proved effective for various weather variables like temperature (Hagedorn et al., 2008), precipitation (Hamill et al., 2008) or wind-speed (Thorarinsdottir and Johnson, 2012) and can be applied using a truncated or censored distribution to account for a constraint to non-negative values."*

36-39 "Although originally developed for Canadian weather and vegetation, it is used in many other regions, e.g., by the European Forest Fire Information System (EFFIS) to provide information on wildfires in the EU and neighboring counties (Giuseppe et al., 2020)": sentence needs to be more orderly and written better; also, author's last name is "Di Giuseppe"

We improved this sentence in the revised manuscript and corrected the author's name.

*"Although originally developed for Canadian weather and vegetation, the FWI system is now used in various regions. For instance, the European Forest Fire Information System (EFFIS) employs the FWI to provide information on wildfires in the EU and neighboring countries (Di Giuseppe et al., 2020). "*

39-40 "One advantage of using FWI is the relatively simple calculation only requiring four weather parameters in addition to information of the season (time of year) and geographical location": Rephrase, e.g. "The main advantage of using FWI is its relatively simple computation, only requiring four weather parameters and information about the season (time of year) and geographical location"

We rephrased this sentence to:

*"The main advantage of the FWI system is its relatively straightforward computation, requiring only four weather parameters and information about the season (time of year) and geographical location as input parameters."*

42-43 "the moisture content of three separate fuel layers of different depth and diameter": this is one interpretation of the three parameters (of course, the main one), meaning that - more or less- they contribute to the fire danger with the same time scale of a certain fuel layer. For example, the DC can also be an index of the lack of precipitation for a long time. I tend to be more cautious when interpreting these indices, but it is a relatively small issue.

We rephrased this section to add the depth of the moisture levels (see comment 15, reviewer 2). We also added a comment about the response of the moisture codes to weather variations, please see reply to the first comment.

48 "Often the FWI is classified": the FWI is often classified

We changed the order of this sentence accordingly.

50 "e.g. vegetation types": since this is the second time it appears, I have to point out that you cannot put "e.g." in the middle of a sentence without it being in parentheses or in a parenthetical expression (between commas). This is not formal enough for a paper, in my opinion.

We corrected this throughout the paper. Thank you for this comment!

52-53 "Fuel moisture codes (FFMC, DMC, DC) and consequently FWI values are dependent on preceding conditions. Thus, the preceding days noon values are used for FWI calculations and the calculations need to be initialized": To be put above, together with the input variables, and to be written more clearly.

We moved the sentence about the initial values to the paragraph about the moisture code calculation (see reply to first comment). The sentence about the dataset used for initial values was moved to the Forecast and observation data section (2.2).

62 "The resolution of the used TIGGE data": The resolution of the TIGGE data used in this paper

We rephrased this sentence accordingly.

65 "can not": cannot

This typo has been fixed.

Figure 1 (caption): please provide a reference (IPCC) for the AR5 regions.

We wrongfully called the regions here AR5 when it was supposed to be the AR6-WGI regions. We corrected that and added the reference for the AR6 regions.

90 Formula (2): shouldn't it be log($\sigma_{kl}$)?

It should indeed be log($\sigma_{kl}$). We apologize for this mistake and correct the formula.

91-92 "The logarithmic link log(sd)": define sd (ensemble standard deviation?). Besides, isn't it $sd_{kl}$ ?

sd is defined as the standard deviation of the 50 ensemble members in the previous sentence. We agree with the reviewer that it should be $sd_{kl}$ and corrected this mistake.

119 "The bias of the forecast can be accessed by simply evaluating the difference between the average forecast and average observation, which is defined as the mean error (ME)":

The bias of the forecast can be accessed by simply evaluating the difference between the average forecast $F_i$ and average observation $O_i$, which is defined as the mean error (ME)

We added the symbols $F_i$ and $O_i$ to the sentence.

137 "defined by the 6th IPCC Assessment Report (AR6 (Iturbide et al., 2020))": did you not show the AR5 regions before (Figure 1a)? It needs to be coherent, at least in the name in the caption of Figure 1a, if the regions did not change.

Thank you for making us aware of this mistake. The reference regions did not change and it is supposed to be AR6. We changed it to be coherent throughout the manuscript.

140 "Other regions can be selected as well, e.g. the calibration can also be done country-wise or at even smaller level.": colloquial, rephrase like "the calibration can also be performed over smaller areas (e.g., single countries)"

We rephrased this sentence to:

*"The calibration can also be performed over smaller geographical areas, e.g., individual countries."*

158 "the forecasted FWI is too low compared to observations": rephrase in a more formal "the forecasted FWI underestimates the observations[…]" or similar.

We rephrased the sentence to:

*"Uncalibrated forecasts have a negative bias for all lead times, indicating that the forecasted FWI values are consistently lower than the observed values."*

**Reply to the comments of reviewer 2**

**Major comments**

1. Study workflow is not very clear, e. g. which datasets are used in which step. This is due to missing research questions and the structure of the manuscript. This should be addressed by:

   a. adding research questions, e. g. can NGR be used to calibrate FWI derived from mid-range weather forecasts, how well does NGR improve the FWI derived from mid-range weather forecast. The research questions should be placed at the end of the introduction, as they help the reader to know what to expect from the following chapters.

Thank you for this comment. We added a sentence stating the research question at the end of the introduction, as follows:

*"In this article, we investigate whether non-homogeneous Gaussian regression (NGR) can be used to calibrate the Fire Weather Index (FWI) derived from medium-range ensemble weather forecasts and assess the extent to which NGR improves the accuracy of the FWI predictions."*

   b. Restructuring the manuscript by summarizing chapters 2 to 4 to a data and methods section with subchapters (the following is a suggestion):
      i. 1. Introduction, providing research questions at the end of the chapter
      ii. 2. Data and Methods,
         1. 2.1 FWI and FWI calculation
         2. 2.2 Forecast and observation data
         3. 2.3. Validation and Calibration methods
            a. 2.3.1 NGR
            b. 2.3.2 Verification methods
      iii. 3. Results
      iv. 4. Discussion (missing, see other comment.)
      v. 5. Conclusion

We agree that the restructuring of the manuscript makes the workflow better understandable and adjusted the structure of the manuscript. However, we decided to integrate the Discussion into the conclusion as "4. Discussion and conclusion". The new structure of the revised manuscript is as follows:

1. Introduction
2. Data and methods
   2.1.      Fire weather index calculation
   2.2.      Forecast and observation data
   2.3.      Calibration and verification methods
      2.3.1.    Non-homogeneous Gaussian regression

2. Introduction: Please add a paragraph why you chose the NGR method over other methods, e.g. other variations of EMOS calibrations or bias-correction methods (see Whan et al. 2021, https://doi.org/10.1016/j.wace.2021.100310).

We added a short paragraph at the end of the introduction mentioning other calibration methods and why we chose NGR.

*"Various methods have been developed for calibrating probabilistic ensemble forecasts. Commonly used calibration methods include Bayesian model averaging (Raftery et al., 2005), non-homogeneous Gaussian regression (Gneiting et al., 2005), logistic regression (Hamill et al., 2004) and non-parametric ensemble post-processing methods such as rank histogram techniques (Hamill and Colucci, 1997), quantile regression (Bremnes, 2004) and ensemble dressing approaches (Roulston and Smith,2003). Non-homogeneous Gaussian regression (NGR) is one of the most commonly used calibration methods and adjusts both ensemble mean and spread, while still be efficient and easy to implement. It has been proved effective for various weather variables like temperature (Hagedorn et al., 2008), precipitation (Hamill et al., 2008) or wind-speed (Thorarinsdottir and Johnson, 2012) and can be applied using a truncated or censored distribution to account for a constraint to non-negative values."*

3. Fire weather index calculation: It is not clear how the FWI is derived. Please clarify in section 2 how you derive the FWI, i.e. which R-package or python package you are using, as there are differences between cffrds and the ECMWF fire products derived from the ECMWF GEFF- Model (see Vitolo et al. 2019, https://doi.org/10.1038/sdata.2019.32)

We agree that it was not very clear how we calculate the FWI. We added a sentence in the end of section 2.1 stating that we have implemented the FWI calculation use Python programming language and following the source code provided by Wang et al. 2015, (https://publications.gc.ca/site/eng/9.805607/publication.html), as follows:

*"For this study, we implemented the calculation of the Canadian Forest Fire Weather Index (FWI) using Python programming language, following the source code provided by Wang et al. (2015), with modifications to utilize gridded input data."*

To check if our implementation provides consistent values with other FWI algorithms, we tested our method using the test values given in Vitolo et al. 2019 and default initial values for FFMC of 85, DMC of 6 and DC of 15 (instead of the ERA5 climatology values used in the presented paper). The outputs closely match those of ECMWF fire products and cffrds,

either exactly matching cffrds outputs or falling between the values of ECMWF fire products and cffrds

| Code Source | FFMC | DMC | DC | ISI | BUI | FWI |
|---|---|---|---|---|---|---|
| R package cffdrs * | 87.69 | 7.29 | 17.76 | 10.85 | 7.28 | 9.46 |
| ECMWF * | 87.70 | 8.54 | 19.01 | 10.80 | 8.49 | 10.10 |
| This study | 87.69 | 7.59 | 18.76 | 10.85 | 7.59 | 9.62 |

* values taken from Vitolo et al. 2019

4. Forecast and observation data:

   a.  Lines 57 – 65: You state that you derive the FWI from the ECMWFs operational forecast system (ENS). Later you state that you use the TIGGE dataset. Can you clarify if you used the ENS dataset, the TIGGE dataset or both datasets later in your analysis? I understand that the TIGGE dataset has a higher temporal resolution than the ENS dataset, however, the spatial resolution is coarser (0.5° vs 0.2°). Later in your results section you show the earliest results for 36h lead time. Can you add a sentence why you are more interested in the increased temporal resolution of the TIGGE dataset over the spatial resolution of the ENS dataset in your manuscript?

The mentioning of ENS and TIGGE dataset is indeed confusing. In this study, we use the ECMWF ensemble forecasts retrieved from the TIGGE database (spatial resolution of 0.5°), although the method is intended for post-processing the operational ECMWF forecast (spatial resolution of 0.2°) to provide FWI forecasts for the SAFERS project. However, we don't have access to multiple years of ECMWF ENS forecast data with higher spatial resolution. To still perform a statistical analysis, we decided to use TIGGE data instead.

To avoid confusion, we decided not to mention ECMWF operational forecasts. Instead, we introduce in the beginning of the chapter 2.2 "Forecast and observation data" all datasets that are used in this study, i.e., the TIGGE ECMWF forecasts, ERA5 reanalysis and High-resolution forecasts.

*"2.2 Forecast and observation data*

*In this study, we use ensemble forecasts from the European Centre for Medium-Range Weather Forecasts (ECMWF) to calculate ensemble forecasts of the FWI. ECMWF medium-range ensemble forecasts consists of 50-members, initialized twice a day at 0000 and 1200 UTC and provide forecasts up to 360 hours (15 days). The forecasts are derived from the TIGGE archive (Bougeault et al., 2010), which provides operational medium-range ensemble forecast data for non-commercial research purposes. The data is accessible through the ECMWF API1. The temporal resolution of the ensemble forecast data used in this paper is 6h for all lead times and the spatial resolution is 0.5° (~50 km). We use only the forecasts*

*initialized at 0000 UTC. Although available forecasts cover the whole globe, our focus is on the European region from 25°N to 72°N and 25°W to 39.80°E and we specifically use forecasts from the years 2021 to 2023.*

*For the FWI calculation, we derive initial values for FFMC, DMC, and DC from ERA5 reanalysis data (Hersbach et al., 2020) to account for preceding conditions at forecast initialization. The initial values are determined using the climatological from 40 years of historical data (1980-2019) for each day of the year, using a 15-day rolling mean centered on each day. ERA5 reanalysis data can be retrieved from the C3S Climate Data Store (CDS) (Hersbach et al., 2017).*

*For calibration and verification purposes, we use ECMWF high-resolution forecasts initialized at 0000 UTC. ECMWF high-resolution forecasts have a spatial resolution of 0.1° ( ~9 km) and a temporal resolution of 1 hour and can therefore give a more accurate picture of the actual weather conditions than medium-range ensemble forecasts with a coarser resolution. The FWI values calculated using ECMWF high-resolution forecasts with the shortest lead time to the local noon with corresponding 24h precipitation act as substitute for actual observations and are hereafter called analysis. Ideally, FWI forecasts would be verified using FWI values calculated using surface observations of the relevant weather parameters as the FWI cannot be observed directly. However, measurement stations that provide continuous observations of all necessary weather parameters are sparse and only yield point-wise verification. Furthermore, for an operational calibration of the FWI, observation data needs to be available rapidly."*

    b.   Lines 65 – 68: These sentences should be moved to the paragraph where you discuss how you verify the FWI from ECMWF data to observation data.

We moved these sentences to fit better in the context of the verification of FWI analysis and observation data. Please see changes to Section 2.2 given in the previous comment.

    c.   Line 68: "We therefore use the FWI calculated ECMWF high-resolution forecasts …". You did not introduce what the ECMWF high-resolution forecasts are yet. You can optimize this by merging this sentence with the next sentence (i. e. "ECMWF high-resolution forecasts have …"), but it remains now unclear why you introduced the ENS and the TIGGE dataset before. Please clarify in this section on which datasets you derive the FWI from and for which later steps you use which datasets (NGR regression and verification).

This section (2.2) has been completely rewritten to make it more clear which dataset is used for which step and to accommodate the changes to Figure 1 suggested by reviewer 1. Please see reply to comment 4a and second major comment of reviewer 1.

5. Results / Figures:

   a.  All figures with subplots: add labels for subplots, i. e., (a), (b), (c) as suggested in the NHESS publication guidelines https://www.natural-hazards-and-earth-system-sciences.net/submission.html#figurestables)

Labels have been added to all subplots to be in accordance with the guidelines.

   b.  In your written text you relate to lead times as days while in the figures you show lead time in hours on the x-axis. Can you synchronize this information? In the current state the reader has to transform between written "7 days" to 7*24h in the x-axis of the respective plot. You could write the hour also in brackets next to the days in the text.

We believe the article is easier to follow using lead times in days instead of hours but agree that using both units throughout the manuscript is confusing. We applied the suggested changes and added the lead time in hours after every mention of the lead time in days. Please note that the lead times in hours do not exactly correspond to the typical conversion of days to hours (e.g., 7 days is here 180 hours instead of 168 hours). This is because FWI forecast are issued for 12 UTC while they are initialized at 00 UTC. The first day's forecast has therefore a lead time of 36 hours and not 24 hours.

6. Terminology "short lead times": You state multiple times that your results work best for short lead times. Can you clarify in your manuscript how you define short lead times (e. g. 72h or 132h) or be more specific which lead time you still find good performing (e. g. rephrasing to something like: "for short lead times up to 84h").

Thank you for this comment. We agree that it is not very clear what is meant with short lead times. We clarified in the manuscript that an improvement is visible for all regions up to a lead time of 84 hours, and we added this information to every instance where we mention that the method works best for short lead times.

7. The discussion is missing. In my opinion the discussion is an integral part of a research paper and as the of the study presents a novel way of calibrating mid-range weather forecasts, it would be good to critically reflect on the results:

   a.  Is the FWI a suitable predictor for fire events in all three regions? What are the challenges regarding wildfire hazard in the three regions? For example, you could address why the postprocessing works particularly well in the MED and summer months of WEU and why not for NEU? Further, you could reflect on how low FWI values, as present in NEU, affect your method.
   b.  How does your method (NGR) compare to other post-processing methods? Select a two to three different studies, with a similar research question and set your results

in a broader context. For example, you could discuss why you are correcting the FWI instead of the input variables of the FWI, why you chose the NGR method and not a bias correction method or machine learning based method (see Whan et al. 2021 https://www.sciencedirect.com/science/article/pii/S2212094721000086) and Worsnop et al. 2021 (https://journals.ametsoc.org/view/journals/wefo/36/6/WAF-D-21-0075.1.xml?alreadyAuthRedirecting)

c.  What can stakeholders take away from your study. You illustrated quite nicely in the introduction that post-processing helps to make accurate forecasts helping first responders. What do you wish this target group takes away from your results, e. g. will more firefighting resources be placed at locations with elevated FWI?

Thanks for this comment. We decided to incorporate the discussion in the conclusion, which reads now as follows:

*"We investigated whether non-homogeneous Gaussian regression (NGR) can be used to calibrate fire weather index (FWI) forecasts based on medium range ensemble weather forecasts by the European Centre for Medium-Range Weather Forecasts ensemble forecasts (ECMWF). To estimate the calibration coefficients of the NGR, we employ a truncated Gaussian distribution with cut-off at zero and use forecast and observation pairs of the last 30 days preceding the forecast. We used ECMWF high-resolution weather forecasts with the shortest possible lead time to calculate the FWI analysis and used these as substitute for observations. Although the FWI analysis seems to underestimate the FWI slightly, a good correlation is observed.*

*FWI forecasts using medium range ensemble weather forecasts perform generally quite well when compared to the analysis. However, calibration improves the forecasts especially at short lead times up to 84 hours. In the Mediterranean region and Central Europe an improvement of FWI forecast with respect to the FWI analysis is also apparent for longer lead times up to 8 to 10 days, respectively. This is likely caused by the generally higher values in those regions and is supported by the monthly averaged metrics in the appendix, which show a stronger improvement caused by the calibration in the months with high FWI values. Hence, it can be concluded that the calibration performs better for higher FWI values, as indicated by the better performance during the summer months in West and Central Europe and in the European Mediterranean. However, the calibration method shows limitations for low FWI values, which could be observed in the NEU reference region and especially for longer lead times. Although it would be ideal for the method to perform effectively across the entire range of FWI values, it is generally more critical that it demonstrates a good performance at higher FWI values, where the potential fire danger is more significant. While long-range forecasts of potential fire danger are valuable, short-range forecasts, especially for the first 1-3 days, are usually more critical for firefighting resource management. Reliable*

*and accurate forecasts, particularly when the fire risk is high and over the short-term time frame, are crucial for decision-makers and emergency responders to effectively coordinate resources. The improvement of FWI forecasts using the presented calibration method improves the ability to anticipate fire danger, ultimately supporting better response management and shows that a relatively simple method can provide good results compared to more complex approaches, e.g., bias correction (Cannon, 2018) or the correction of the input parameters instead of the FWI as in Worsnop et al. (2021). When correcting the individual input parameters, different models need to be applied for the individual variables, which require careful verification and access to quality controlled observation over the whole study region. To further improve the presented calibration method for fire weather index forecasts, it could be tested if calibration of individual components of the FWI system e.g. FFMC, DMC and DC would improve overall skill of the forecast. Furthermore, more advanced models using additional predictors, e.g. elevation or land-use, could improve the calibration but were not tested here."*

The take-away message of our study is that NGR can improve FWI forecasts and therefore provide more accurate predictions of potential fire danger. In addition to being illustrated in the introduction, we again mention this now in the conclusion, also. We refrain from giving suggestions for possible actions based on the provided results, e.g. making suggestions for firefighting resources, as this is not the purpose of this article. However, supporting firefighting resources is the objective of the SAFERS EU project, which is mentioned in the introduction.

**Minor comments**

1.  Line 6 & Line9: you use the terms post-processing and calibration interchangeable, please choose one term.

    Thanks for pointing this out. We decided to use the term calibration and changed this throughput the manuscript.

2.  Lines 9 – 11: Be more specific about what you mean by short lead times (e. g. 84h?) and regions with elevated FWI (e. g. MEU)

    We included more specific information about the lead time and region as follows:

    "The calibration improves FWI forecast particularly at shorter lead times up to 84 hours and in regions with elevated FWI values, i.e., areas with a higher wildfire risk such as Central and Mediterranean Europe. "

3. Lines 13: I would drop the word "recent" and replace "wildfire in Greece 2023" by "wildfire season of 2023".

We changed the sentence according to the suggestion.

4. Line 15: Drop "Also" at beginning of sentence.

We rephrased this sentence to avoid starting with "Also".

5. Line 16: Drop "But" or make this sentence sound more formal.

We rephrased the whole sentence in order to sound more formal and to accommodate the added references (comment 6).

"While the Mediterranean region continues to face the highest occurrence of wildfires, Central and Northern Europe have experienced an increase in extreme temperature events and heatwaves in recent years (Ibebuchi and Abu, 2023; Rousi et al., 2023; Ionita et al., 2017; Barriopedro et al., 2011). Extended warm and dry periods raise the fire danger and may cause wildfires in regions that were previously not considered wildfire hotspots (San-Miguel-Ayanz et al., 2021; De Rigo et al., 2017). Examples are the 2018 heatwave, which caused wildfires in Sweden (San-Miguel-Ayanz et al., 2019) and across the United Kingdom (Sibley, 2019) or the wildfire outbreaks in Germany and the Czech Republic during the summer of 2022 (Skacel et al., 2022)."

6. Line 17: Provide references for your statement.

References for the increased extreme temperatures and heatwaves have been added to the statement (see comment 5).

7. Line 19: Provide references for your statement.

References are added (see comment 5).

8. Line 20: Here it would be great if more than one example could be provided, e. g. one for each subregion.

We added two more examples, one of which is for the WCE subregion (see comment 5).

9. Lines 22 – 25: Switch the order of the sentence to stress more clearly that weather forecast is part of SAFERS or drop mentioning the project.

We changed the order of the sentence to make it more clear that the weather forecasts are part of SAFERS.

10. Line 26: Add Di Giuseppe et al. 2016 (https://doi.org/10.1175/JAMC-D-15-0297.1) as a reference.

   The suggested reference has been added to this sentence.

11. Line 26: Watch out that your citation tool takes the names correctly. It is van Wagner and Di Giuseppe not Wagner and Giuseppe.

   Thank you for this comment. This was a mistake in the used Bibtex reference. We checked and corrected all author names with surname prefixes.

12. End of line 27: Here I am missing a short explanation of what is the difference between deterministic and probabilistic weather forecast. A short explanation would be helpful to emphasize that the topic of the manuscript is postprocessing probabilistic forecasts.

   We added a short explanation of the difference between deterministic and probabilistic forecasts.

   "While deterministic forecasts provide a single forecast based on a given set of initial conditions, probabilistic ensemble forecasts offer a range of possible outcomes by using slightly perturbed initial conditions, giving a more comprehensive picture of potential weather conditions and providing an estimate of the forecast uncertainty."

13. Line 28: drop the word "may" or provide a clear statement whether post-processing is needed. Consider also my first comment on your interchangeable usage of post-processing and calibration. This confuses the reader.

   We dropped the word "may" and changed post-processing to calibration.

14. Line 39: Chose a more scientific formulation than "is a relative simple calculation" for the FWI.

   We changed the formulation of this sentence to "*straightforward computation*".

15. Line 43: Add the depth of the moisture levels.

   The depth and type of fuel layer have been added. All changes to the FWI calculation section (2.1) can be seen in reviewer 1, comment 1.

16. Line 48: Rephrase the sentence starting with often, e. g. "The FWI can be classified"

   The sentence has been rephrased to:

*"The FWI is often classified into danger classes and values above 50 are considered extreme."*

17. Lines 52 – 54: I would change the order of the sentences to make the statement at the beginning of the paragraph clearer, e. g. "we use climatological mean values …, to account for preceding conditions at the initialization".

    *We follow you suggestion and changed the order of sentences.*

18. Lines 59, 63 and 70: Please provide an approximation of the grid resolution in km in brackets?

    *An approximation of the grid resolution in km has been added in those locations.*

19. Line 60: it should be "derived from the TIGGE archive".

    *The article has been added.*

20. Line 61 and following: Please add dataset after TIGGE, i. e., "The TIGGE dataset …".

    *We completely rewrote the forecast and observation data section (2.2) and don't refer to the TIGGE dataset in those sentences anymore.*

21. Line 62: Please add "the" to ECMWF API.

    *We added the article "the".*

22. Line 62: Please rephrase sentence to "the temporal resolution of the TIGGE dataset …".

    *We rephrased this sentence to:*

    *"The temporal resolution of the ensemble forecast data used in this paper is 6h for all lead times and the spatial resolution is 0.5° (~50 km)"*

23. Lines 65 – 78: Please rephrase this paragraph. Keep the statement that the FWI has multiple input variables. Explain for which later steps you use which dataset to calculate the FWI. Mentioning the station data here is confusing.

    *We rephrased the whole section 2.2 (see major comment 4) and hope it now clear which dataset is used for which step. The station data is now mentioned in the second part of this section after introducing all forecast datasets.*

24. Line 71: I am not sure which dataset you are meaning by "those", please clarify.

Section 2.2 has been completely rewritten and it should now be clear which dataset is meant.

25. Line 75: How many of the 682 stations are in Finland and how many are outside of Finland. Can you provide values.

Because the correlation coefficient of FWI analysis and FWI observation is basically identical for Finland and the stations outside Finland and no additional conclusions can be drawn by separating Finland from the rest of the stations, we decided to merge the dataset. It is therefore no longer necessary to mention the number of stations in Finland separately.

26. Fig 1 / Line 74 (first mentioned):
    o Add letters for subfigures.
    o Fig 1a
        ▪ Use a different projection, e. g. Lambert Conformal Conic, as the northern latitudes are strongly distorted.
        ▪ It would be very nice to have the stations shown in Fig 1c on the map of Fig 1a as well.
        ▪ Place the region legend (i.e., NEU, WCE, EUMED) inside Fig 1a.
        ▪ Place the legend of the countries (i.e., Finland and others) at a position, where it is clear the legend belongs to both Figures (i. e. Fig 1a and Fig1b)

We implemented all suggested changes in Fig. 1a.

[Figure]

*Fig.1*

    o Fig 1b:
        ▪ Provide a legend for the regression line. Is this the line for all stations, or for only "other" stations (outside of Finland) or for only Finland stations?
        ▪ Can you provide a line for Finland as well?

There were 2 regression lines in Fig 1b, one for Finland and one for the other stations. Those regression lines were basically identical and therefore not very well visible. Figure 1b has completely been changed according to suggestions of reviewer 1 (major comment) and we now show a separate Figure (Fig.2) with histograms of the FWI values and correlation coefficients.

[Figure]

Fig.2

- o Fig 1c:
    - Please show a 3rd station for WCE.
    - Add the location of the stations in Fig 1a.
    - Why are there missing values in the Greece station in the winter of 2022 and 2023? You previously stated that all your selected stations have a sufficient data coverage. Please clarify that your consecutive 200 days refer to the summer half (?) in the Figure caption.

We added a third station (Meiningen, Germany) as an example for WCE and marked its location on Fig. 1a. Accessing data of stations outside Finland with year-round continuous data proved challenging. To expand our analysis to stations outside of Finland, we included stations with some missing data, as long as they had at least 200 continuous days of data. This period isn't restricted to summer months. We chose this 200-day threshold to avoid too frequent reinitialization of the FWI calculation, which could affect the comparison of observation and analysis.

27. Line 77: Provide a reference to Fig 1b as your statements originate from the figure.

    Fig 1b has been updated and mentioned scatterplot is no longer shown in the updated manuscript.

28. Line 77 and Fig 1b: Is your, I assume linearly derived correlation, mainly driven by the large number of low FWI values? Fig 1c shows quite apparent that for high FWI

values the underestimation is much stronger pronounced than for low FWI values. This would be also a good point to be discussed in the discussion.

The correlation is indeed mainly driven by the large amount of low FWI values, which is one reason we decided not to present the scatterplot and linear correlation in the revised manuscript. Instead, we show the distribution of FWI values calculated from observation and analysis using histograms (Fig. 2a,b) and the distribution of the linear correlation coefficient in Fig.2c. Furthermore, we point out that especially for high FWI values the analysis is underestimating the FWI with regard to the observation when referring to Fig.1 b,c,d.

29. Line 79: Provide a reference to Fig 1c.

Fig. 1c has been updated to Fig.1 b,c,d. References to the respective figure are added in the revised manuscript.

30. Line 80: Please clarify which datasets you use for longer forecasts.

We use the medium-range ECMWF ensemble forecast and clarified this in the revised manuscript.

31. Line 83: Please clarify which dataset(s) you mean by the FWI forecasts.

We added "*medium-range*" to the FWI forecasts to clarify which forecasts we mean. With the changes made in Section 2.2, it should now be clearer which dataset we refer to.

32. Line 86: Please specify what you are calibrating, e. g. the parameters of the NGR or the whole post-processing pipeline.

We added that we are calibrating the FWI forecasts to this sentence.

33. Line 86: It should be: "the FWI".

We added the article.

34. Line 91: Please clarify from which dataset the 51 ensemble members come, e. g. by adding (ENS) in brackets. You describe the statistical part very clearly, but it is not clear to which datasets you are applying the formulas.

The ECMWF ENS forecast is no longer mentioned in the revised manuscript. With the changes made to Sect.2.3.1, it should be evident that the calibration is done for the FWI calculated from the ECMWF forecasts (derived from TIGGE database).

35. Line 96: Please specify what the training area is.

In this study the AR6 WGI reference regions are used as training areas. We agree that the training areas need to be introduced already when explaining the NGR method and therefore moved the introduction of the AR6 WGI reference regions from the results section to Sect. 2.3.1.

*"The training data is defined using a specific training area and a 30-day rolling window preceding the forecast as training period. As training areas, we use here climatic reference regions, defined by the 6th IPCC Assessment Report (AR6-WGI (Iturbide et al., 2020)), which divide the European domain into Northern Europe (NEU), West and Central Europe (WCE), and the Mediterranean (MED). In this study, however, we use only the European part north of the Mediterranean Sea, referred to hereafter as the European Mediterranean (EUMED). The reference regions are marked grey in Fig.1(a). The calibration can also be performed over smaller geographical areas, e.g., individual countries."*

36. Line 97: Why do you switch terminology from NGR to EMOS, I understand that this is the approach, but it would be good to decide for one name.

    We removed the reference to EMOS to avoid confusion.

37. Lines 98 -100: Can you provide results, e. g. a table or a plot, for these findings. This could be part of your supplementary material.

    We added an additional figure in the Supplementary material Fig. S1. This figure shows the average CRPS for a test time period (June-July 2021) using different training areas.

38. Line 100: Here it becomes apparent, that it is not clear what the training area and hence smaller geographical training areas should be. Please clarify and provide results in the supplementary material.

    In the revised manuscript, we now introduce the concept of training areas prior to this statement (see comment 35) and clarify that smaller geographical regions can also be used for calibration purposes. We hope this revision makes the meaning of training areas clear.

39. Line 114: Please add a note that you call RMSE later spread and skill metric (i. e. line 122 and line 144).

    This has been clarified in the text.

40. Line 142: please clarify that you compare fire season length of Northern Europe to Southern Europe.

We added the note that we compare Northern and Southern Europe here.

*"The main fire season in Europe is typically from May until October but varies strongly in length and intensity across regions, e.g. the fire season starts later and is shorter in Northern Europe compared to Southern Europe (San-Miguel-Ayanz et al., 2012)."*

41. Line 145: the grid points "within" rather than "of" the three study areas.

    We changed the wording accordingly.

42. Line 146: I can't follow how you derived the RSME of the climatology. Can you describe this briefly.

    We added a brief explanation how the climatology RMSE is calculated as follows:

    *"The RMSE of the climatology, shown by the solid blue line, is calculated similarly to the forecast RMSE. However, it uses the climatology derived from 40 years of ERA5 reanalysis data (see Sec.2.2) for each day of the year instead of using the forecast."*

43. Line 151, Line 163: clarify that your calibration is done as a post-processing.

    This is now clarified in the first sentence of the results section:

    *"In this section, we present results of post-processing FWI forecasts of the years 2021 to 2023 by applying the introduced calibration method."*

44. Line 152: Specify what you mean by short lead times, e. g. 132h?

    In this specific case, we decided to drop the part of the sentence mentioning short lead times. Short lead times refer only to the NEU region (~132h) while the RMSE of the calibrated forecast is still smaller than the RMSE of the raw forecasts up to 228h for WCE and 300h for EUMED. These lead times are mentioned in the subsequent sentences.

45. Line 154: Provide lead time in hours in brackets after "7 days", i. e. 7 days (168 h).

    We included the lead times in hours in brackets after every following mention of the lead time in days. It has to be noted that the lead time in hours doesn't exactly match the time in days because the FWI forecast is done for 12 UTC using the forecast initiated at 00 UTC. The first day's forecast has therefore a lead time of 36 hours and not 24 hours.

46. Figure 2:
    o   Add labels (letters a, b, c) to subplots.

<li>○ Place legend outside of NEU to make it clear it belongs to all three subplots.</li>
<li>○ Adjust the y-axis label (Spread/ RMSE) to your figure caption, which is currently "spread and skill". I would expect them to be the same, e. g. spread and RSME in the figure caption or spread and skill in the y-axis label.</li>

All suggestions have been applied to Fig.2 (now Fig.3). We adjusted it to be Spread/RMSE on caption and y-axis label.

[Figure]

*Fig.3*

47. Line 157: Rather the three subregions than the respective area.

   The sentence has been rephrased accordingly.

48. Line 158: Rephrase "too low" to something like "lower than observations".

   We rephrased the sentence to:

   *"Uncalibrated forecasts have a negative bias for all lead times, indicating that the forecasted FWI values are consistently lower than the observed values."*

49. Line 160: "The improvement" instead of "This improvement".

   We changed the article accordingly.

50. Line 164: Provide numbers for what you think is slightly positive.

   We provide numbers of the positive mean error now.

   "In Northern and Central Europe, the bias is slightly positive after calibration especially for longer lead times, ranging from less than 0.1 to 0.4."

51. Line 167: specify short lead times.

   We specified here that lead times up to 84 hours are considered short lead times.

52. Line 166 – 169: This finding would be a good point to discuss in the discussion. For example, I would be interested what these findings imply for the application of your suggested post-processing technique.

   This has been added to the discussion in "4. Discussion and conclusion", see major comment 7.

53. Figure 3: please add letters to subregions.

   The abbreviation of the subregions has been added to the caption.

54. Figure 4: Please add letters to subregions.

   The abbreviation of the subregions has been added to the caption.

55. Figure 4 caption: drop "the grid of".

   "the grid of" has been dropped.

56. Lines 171 – 172: Can you discuss this in your discussion section? Does your approach perform well for higher FWI values, suggested by the better performance in WEU in July and August and MED? Does your approach need to perform well on or low no-fire danger days?

   This has been added to the discussion in "4. Discussion and conclusion", see major comment 7.

57. Figure 5:
   o   Add labels to the subplots (i. e., a, b, c, d)
   o   Plot the land-sea boundary to improve the visualization.
   o   Drop the large white space in the south and west of the plot in such a manner that the plot is filled with results.

   All suggestions to improve the visualization of Figure 5 (now Figure 6) have been implemented, additionally we changed the map projection to be in accordance with the map in Figure 1.

[Figure]

*Fig.6*

58. Line 176: Here you mention the first time that you calibrate the coefficients of the NGR, this is not clear in your previous description of the post-processing method. Please clarify this in the method section. In Line 176 add "of the NGR" after "to estimate the calibration coefficients".

   We added "of the NGR" to the sentence. The method section has been improved and is hopefully clearer now.

59. Line 177 - 179: This sentence belongs to the discussion section and not the conclusion session. Also, I suggest adding more meaning to this sentence, e. g. what are the implications of sparsely available data?

   We removed this sentence from the conclusion.

60. Line 180: Drop "Thus".

   "Thus" has been dropped.

61. Line 180: which dataset do you mean by high-resolution weather forecast with short lead time? Is this the third dataset you introduced in the data section?

Yes, the high-resolution forecasts are the third dataset that is introduced. We restructured the Forecast and observation data section (now Sec. 2.2.) and hope it is now clear what we mean with high-resolution data throughout the manuscript.

62. Line 181: Make clear that you mean the dataset "analysis" and not the analysis of the FWI.

We added a reference to the previous sentence to make clear that we mean the FWI calculated by the forecast when referring to analysis.

63. Line 189: At the end of your conclusion, I would expect a last sentence coming back to your initial statement that this is important for fire resource management and the SAFERS project. Please add such a sentence.

We added a sentence to the conclusion that highlights the significance of our findings for fire resource management as follows:

"Reliable and accurate forecasts, particularly when the fire risk is high and over the short-term time frame, are crucial for decision-makers and emergency responders to effectively coordinate resources. The improvement of FWI forecasts using the presented calibration method improves the ability to anticipate fire danger, ultimately supporting better response management."

64. Line 209: Add "Di" to "Di Giuseppe".

We fixed this mistake in the reference.

65. Line 249: Add "Van" to "Van Wagner".

We fixed this mistake in the reference.

**Reply to second review comments**

**Thank you once again for providing valuable comments to further improve the paper. Please find our replies to your comments below and the changes to the manuscript marked in the manuscript.**

**General minor comments**

1. Terminology

    a. Be consistent in how you refer to figures in your text (see comment 22): You use Figure 2a (line 107), while in line 99 you use Fig. 1 and 2. Also make sure the blank space is consistent between figure and figure number as well.

    We use the abbreviation "Fig." whenever we refer to a figure in the running text and "Figure" when the reference to the figure appears in the beginning of a sentence, which is in accordance with the NHRESS manuscript guidelines. We checked and corrected the blank spaces between the "Fig." and the numbers.

    b. Name your metrics consistently, e. g. RSME and skill are used interchangeably. pointed it out in comments 35,39 and 41 but there might be locations I oversaw. Please check this carefully before resubmission.

    We agree that the use of RMSE and skill should be consistent. We replaced "skill" with RMSE whenever discussing the calculated RMSE, but we continue to refer to "skill" when referring to the performance of the method, as suggested.

2. Definitions:

    a. You use relative statements throughout your paper. Make clear what are low / high FWI values or what is a good correlation. Add numbers to these relative terms and state why you use these numbers as thresholds to define a FWI to be high or low, or a correlation to be good (see comments 20 and 24)

    Thanks for the comment. We improved the definition of high/low FWI values by adding in the Fire weather index calculation (2.1) the definition of low FWI values, additionally we added figures to the supplement showing the mean and 95th percentile of the FWI for each month for all regions. Furthermore, we checked throughout the paper that it is clear what we mean with high and low FWI values.

We corrected the statements about the correlation coefficient, please see comment 20 and 24.

b. Same for short- and long-term forecasts. Define these terms in your text, e. g. short term (up to 3 days) and long term (more than 3 days).

We now avoid using the term "long-term forecasts" as it can be confused with the definition of long-term forecasts by ECMWF, which is multiple months. Instead, we give the time range we are referring to (1-2 weeks). Furthermore, we now define short-term forecasts as forecasts up to 3 days.

3. General comment on verification and Fig. 2b:

You state several times that your approach does not work well for small/ low FWI values. What are the thresholds for you to consider FWI to be low? From your study I read that this is below 1, but from the EFFIS danger classes (https://forest-fire.emergency.copernicus.eu/about-effis/technical-background/fire-danger-forecast) low FWI values are considered to be below 11.2. Your underestimation of the FWI in the analysis mainly affects this relevant scale (10 -100) (see Fig 2b).

Thanks for this comment. It is indeed not clear what we mean with low values. In general, we mean low values of the FWI below 10. However, in the description of Fig. 2 we especially look at very low values (below 1). We tried to clarify this by adding in the description of the FWI classes, the definition of low FWI by EFFIS and changed the description of Fig.2 to "very low values".

- Can you mention this in the description of the figure (i.e. line 110)

  We added a comment that the underestimation effects the most relevant scale for wildfire risk assessment, as:

  *"The analysis tends to overestimate FWI values below 6 and underestimate values above 6. Notably, these higher values (above 10) are the most relevant for assessing wildfire danger."*

- Discuss and state stronger that your calibration does not address this discrepancy since you calibrate on the (orange) analysis and not on the (grey) observations density?

  We improved the discussion in chapter 2.1 and hope this is now more clear.

- What are the impactions of your decision to use the analysis as observations in the calibration process. Don't you systematically underestimate large/relevant FWI (FWI > 10) values by calibrating to the analysis over observations? Please add your elaboration on this to the discussion and conclusion.

  It is possible that we underestimate large FWI, however it is difficult to quantify the discrepancy and correct for it because the available stations are not homogenously distributed throughout the study area. Over half of the stations that were available are in Austria, Switzerland, Romania, and Norway (as can be seen in the high density of stations in Fig.1a.). Those regions have a complex topography which can cause discrepancies between model forecasts and observations. Furthermore, there are not many available stations in southern Europe were the FWI is generally high. This also shows the difficulties when using observation data for verification and calibration, the availability and representatitivity of observations for large areas needs to be considered and the quality of the observations, i.e. wind and precipitation, which can strongly vary, needs to be assessed. We added a discussion of this issue to chapter 2.2 and the Discussion and conclusion chapter.

**Detailed minor comments**

1. Line 2: "makes accurate wildfire risk estimation crucial" add a target group, e. g. decision makers, emergency responders, etc.

   We added target groups by changing the sentence to:

   *"Wildfires are increasing in frequency and severity across Europe, which makes accurate wildfire risk estimation crucial for decision makers and emergency responders."*

2. Line 6: You already introduced the abbreviation FWI for the Canadian Fire Weather Index. I suggest to not spell it out here and just use FWI.

   We applied this change and do not spell out the FWI here.

3. Add a strong last sentence to your abstract, in which you point out what the target group and the impacts of your study should be.

   We added a concluding sentence to the abstract.

4. Lines 13 - 21: You added good references for your description of the more northern parts in the second part of your paragraph (line 16ff.). Could you underline your statements in the first part as well?

   *We added two more references (Turco et al., 2019; Rodrigues et al., 2023) for the wildfires in 2017 and 2022 in South Europe.*

5. Line 27: In your abstract you say you look into forecasts up to 15 days which is 2 weeks, while here you state you are interested into "forecasts ranging from a few days to several weeks". Do you refer to your study or the outline of the SAFERS project here and what are several weeks?

   *We here refer to the SAFERS project which provides weather forecasts up to 45 days (~6 weeks), however in this study we are looking only into forecasts up to 2 weeks. We clarified this by replacing "several weeks" with "six weeks" and adding to the next sentence that we are looking into forecasts up to 2 weeks in this paper.*

   *"An important component of SAFERS for identifying high wildfire risk areas are accurate and reliable weather forecasts, ranging from a few days up to six weeks. In this paper, we use the Canadian Forest Fire Weather Index (FWI, Van Wagner (1987); Di Giuseppe et al. (2016)), a widely recognized numerical indicator for forest fire risk, to derive fire risk from weather forecasts with a lead time up to two weeks."*

6. Line 28: Rephrase the sentence and add what you use the FWI for, e. g. "in this paper, we use the FWI ... for deriving fire risk from weather forecasts".

   *We rephrased the sentence to:*

   *"In this paper, we use the Canadian Forest Fire Weather Index (FWI, Van Wagner (1987); Di Giuseppe et al. (2016)), a widely recognized numerical indicator for forest fire risk, to derive fire risk from weather forecasts."*

7. Line 28ff: Here you could improve the reading flow by two things:
   - You should shortly mention the parameters of the FWI, i.e. naming the four weather parameters in it (temperature, precipitation, wind speed and rel. humidity).
   - Add a transition to weather forecasts by adding a statement that these parameters are available in deterministic and probabilistic forecasts.

- o Then you can continue with explaining the pros and cons of deterministic vs. probabilistic forecasts.

We changed this section according to your suggestion as follows:

*"The calculation of the FWI only requires four weather parameters: temperature, relative humidity, wind speed, and 24-hour accumulated precipitation, which are often available from deterministic or probabilistic weather forecasts. While deterministic forecasts provide a single forecast based on a given set of initial conditions, probabilistic ensemble forecasts offer a range of possible outcomes by using slightly perturbed initial conditions, giving a more comprehensive picture of potential weather conditions, and providing an estimate of the forecast uncertainty."*

8. Line 35ff: Thank you for adding this paragraph about other methods and adding your purpose of the study. This section is much clearer now.

9. Line 49: Add a reference after various regions, e. g. Di Giuseppe et al. (2016)

    We added the suggested reference and an additional reference referencing the adaptation of the FWI in Southeast Asia.

10. Line 51: Please rephrase "relatively straightforward computation" to a more scientific language. I disagree that the many empirical formulations of the FWI and its subindices are straightforward to develop, though they are straightforward to apply once a source code is available.

    We dropped this part of the sentence and now only mention what is necessary to calculate the FWI.

11. Line 55: Add ", i.e. Drought Code (DC), Drought Moisture Code (DMC), and Fine Fuel Moisture Code (FFMC)".

    We added the moisture codes as suggested.

12. Line 65: Add ", i.e. Initial Spread Index (ISI)."

    We changed the sentence to:

    *"In the second step, FFMC and the 10-meter wind speed are used to model the potential rate of fire spread, i.e. Initial Spread Index (ISI)."*

13. Line 70: Add reference to EFFIS fire danger classes.

We added a reference.

14. Line 70-71: I like that you reflect on the fire danger level thresholds here, but you do not show results for FWI values later in your results section and you do not group your FWI values into these classes. Therefore, you can remove these two sentences about the fire danger levels.

We would like to keep these sentences as they give some background information about the usual magnitude the FWI. Furthermore, we added what is considered a low FWI according to EFFIS in order to make it better understandable what we mean with low FWI later in the paper.

15. Line 74: Drop "with modifications to utilize gridded input data" or provide the code in a GitHub repository, because I am curious to see/ use it now.

We dropped the sentence because we might not be able to provide the GitHub repository link in time before the publishing of the article.

16. Line 86: You missed a word ("mean"?) after climate.

Thank you for finding this mistake. We added "mean" after climatological.

17. Line 87: Rephrase to "using a centered 15-day rolling mean on each day".

We rephrased the sentence according to your suggestion.

18. Line 89: Here, I do not understand where the ECMWF high-resolution forecast comes from. Is this part of the TIGGE archive or from a different archive? Please clarify.

The high-resolution forecasts are retrieved from ECMWF's Meteorological Archival and Retrieval System (MARS) and not part of the TIGGE archive. We clarified this by adding the reference to MARS.

19. Line 93ff: This sentence is confusing. Aren't you showing weather station data in Fig. 1a)? Please restructure this in accordance with the beginning of your new paragraph and make this a uniform/ clear statement.

Yes, we are showing the FWI calculated from weather station data in Fig.1, but this observation data is not used for the verification. The purpose of showing observation data in Fig.1 is to compare analysis and observation and to justify

that we are using the analysis instead of actual observations. However, we restructured this section in an effort to make it clearer:

"For calibration and verification purposes, we use ECMWF high-resolution deterministic forecasts initialized at 0000 UTC, which are available from ECMWF's Meteorological Archival and Retrieval System (MARS2). ECMWF high-resolution forecasts have a spatial resolution of 0.1° (~9 km) and a temporal resolution of 1 hour and can therefore give a more accurate picture of the actual weather conditions than medium-range ensemble forecasts with a coarser resolution. Ideally, the FWI forecasts would be verified using FWI values calculated from surface observations of the relevant weather parameters, since the FWI cannot be directly observed. However, measurement stations that provide continuous observations of all necessary weather parameters are sparse and only yield point-wise verification. Furthermore, for an operational calibration of the FWI, observation data would need to be rapidly available. We therefore use the FWI calculated using ECMWF high-resolution forecasts with the shortest lead time to the local noon with corresponding 24h precipitation as substitute for actual observations. These FWI values are hereafter called analysis."

20. Line 103: Please put your correlation coefficient (r=0.72) in brackets here. The definition of a good correlation is varying in different science domains, and I would rather read that you found a correlation coefficient of 0.72 and why you think this is sufficient for your analysis.

In this section, we do not refer yet to the correlation coefficient but rather to the agreement of analysis and observation in the time series plots. The correlation coefficient of forecasted and observation FWI and the mean value using all the stations is only given later in this section when referring to Fig.2. We therefore don't think it would be correct to provide the correlation coefficient at this point. However, we replaced "correlation" with "agreement" to make it more clear that we talk about the time series plots and not the correlation coefficient.

21. Line 107: Missing word after Figure 2a.

We added the word "shows" to the sentence.

22. Line 107 and wherever you refer to a figure, here you use Figure 2a, while in line 99 you use Fig. 1 and 2. Please make this uniform across your manuscript.

We use the abbreviation "Fig." whenever we refer to a figure in the running text and "Figure" when the reference to the figure appears in the beginning of a sentence, which is in accordance with the NHRESS manuscript guidelines.

23. Line 112: Do you mean analysis by "forecast-derived"? Please stick to one term across your manuscript.

We replaced "forecast-derived" with "analysis".

24. Line 115: Why is it here only a "fairly good" correlation while it was a "good correlation" in Line103?

We removed "fairly".

25. Line 119: Make the statements about the markers more specific, e. g. "the bright red colored markers".

We changed this statement to "as indicated by the yellow and red coloured markers".

26. Fig. 1b-c legend + Fig 2 legend: shorten "calc. from observation" to just "observations".

We changed this in the figure legends to "observation" as suggested.

27. Fig. 1 caption:
   o "See Fig. 2" to "see Fig. 2c".
   o Add "(b – d)" after example stations.
   o Move "(b)" in front of "in NEU".

We applied all your suggested changes.

28. Fig 2c: Can you add the rmean as a vertical line to the plot?

We added the r_mean as vertical line in Fig. 2c.

29. Line 143 – 146. These two sentences need to be restructured. Mention the length of your training periods in the beginning, e. g. "training periods between 15 to 40 days were tested", then add your benefits and disadvantages of different training lengths.

We rearranged the sentences and now mention the tested training periods in the beginning of these sentence as suggested.

30. Line 147 – 148: What training period are you using in your analysis? You say you found 30 days appropriate for small geographical areas and then that you adopt a regional approach, which is in my understanding a large geographical area. Please clarify which training period you stick with here explicitly.

With the regional approach, we mean that we use all the grid point in the area to derive the calibration coefficient, as opposed to the local approach where coefficients are derived for every single grid point. So, the regional approach is used for small as well as for large geographical areas.

We added a sentence mentioning that we use a 30-day window.

31. Line 158: Please replace several with the verification metrics you are using.

We added the verification metrics that are used to the sentence.

32. Line 160: Drop introductory sentence.

We believe this sentence is necessary to understand why we use RMSE (as measure for the skill) and the spread. However, we rephrased the sentence and added a reference to make it better understandable that we are referring to the spread-skill relationship here.

33. Line 164/ 165: I don't understand how you derive the ensemble spread. Is the ensemble spread the standard deviation (square root of the ensemble variance) of the ensemble? What is the average ensemble variance? The mean of the variance of all ensemble members? Please rewrite this.

The ensemble spread is calculated as the square root of the average ensemble variance, where the average ensemble variance is the mean of the variances of all ensemble members. According to the given reference (Fortin et al., 2014), the square root of the average ensemble variance needs to be used instead of the average of ensemble standard deviation, because smaller values would be achieved unless the spread is the same for all time steps. We changed the sentence to:

*"The ensemble spread is calculated as the square root of the average ensemble variance, where the average ensemble variance is the mean of the variances of all ensemble members (Fortin et al. , 2014)."*

34. Lines 166/ 167: Please be clear in your terminology. Is skill in line 167 the same as RMSE in line 166? Please be consistent with these terms throughout your manuscript and figures, i.e. Fig. 3.

We changed skill to RMSE in this location because RMSE is here the same as skill.

35. Line 185: Replace "introduced calibration method" with "the NGR method".

We replaced this part of the sentence accordingly.

36. Line 186: Name the regions again, i.e. NEU, WCE and EUMED.

We added here the region names again:

*"We use here the AR6-WGI reference regions, NEU, WCE and EUMED, introduced in Sec. 2.3.1 and shown in Fig. 1a."*

37. Line 190: What is a considerably high FWI? Please find a reference using this fire seasons to refer to your Fig 1b-d.

We added the reference to Fig. 1b-d, which shows the higher FWI during the months May-October. Furthermore, we changed the sentence to make it more clear what we mean with the higher FWI values during these months:

*"For the calibration verification, we therefore only focus on forecasts during the months May to October, when the FWI in all regions is substantially higher compared to off-season months (see Fig. 1b-d)"*

38. Line 197-199: Here it becomes apparent that you should be consistent with naming it either skill or RMSE.

We changed skill here to RMSE.

39. Line 200: Move the statement that your method improves the forecast to the beginning of the sentence to make your statement stronger.

We changed the sentence as suggested.

40. Line 202/ 203: Consistent naming of RMSE and skill becomes apparent again. From the reading flow it might make more sense to stick to RMSE when you talk about the measured RMSE (line 202: "skill of raw and calibrated") and use skill when you talk about the performance of your calibration method (line 203: "the forecast lacks skill").

We agree and changed it like suggested to RMSE whenever we talked about the calculated RMSE and skill when referring to the performance of the method.

41. Line 205: What do you mean by large uncertainty? If you mean that the signal-to-noise ratio in NEU is higher than in other regions, because the FWI values are lower

and more unlikely to exceed the variability range, then bring this statement more to the point. You could underline this statement by whatever you mean with "(not shown here)".

Yes, this is what we mean. We changed the statement and added a reference to the figure showing the FWI values in the supplement.

42. Line 206: What are you not showing here and where do you show the relative uncertainty in different regions?! Drop this statement, refer to supplementary material or refer to a Figure.

We dropped this statement.

43. Line 214: Just add in brackets the reference to the figures in your supplementary material.

We changed the reference to the figures like suggested.

44. Line 222: What is your reference in Fig. 5 for stating that the skill worsens in NEU but not in WCE? In Fig 5. the median in WCE goes below 0 at lead time between 180 and 220 and therefore, I think that the skill worsens here as well. In EUMED the skill is not worsening if you refer to the median of the boxplots.

You are right, we corrected these statements and now refer to the median of the CRPSS.

45. Line 223/ 224: I don't understand the context of these two statements: Is this worsening of skill caused by small FWI values or by longer lead times? Didn't you say earlier you excluded FWI values below 1? Can you refer to a figure why you think low FWI values are the cause of the worsening? Throughout your manuscript you bring up this argument frequently, but you never show how high/ low the FWI values in the different regions are, and why you come to this conclusion.

We believe it is caused by the low values and the lower variability of values. In NEU, the FWI values are usually quite small (FWI < 10) and the calibration has less effect, in contrast to EUMED where the range of FWI values is larger. To illustrate what high and low values are in the different regions, we included a figure in the supplement showing the mean and 95$^{th}$ percentile of the FWI for all regions and each month.

46. Line 225: Merge this with the paragraph above. Shorten this by dropping the first sentence and adding the second sentence to the paragraph above by referring to the figure in brackets: "With increasing lead time the skill worsens especially in

mountainous areas in Scandinavia and the Alps, which are also the regions with generally low values throughout the fire season (see. Figure 6)."

We applied this change.

47. Figure 6 comment: You claim that your method does not perform well over mountain regions. Which is true for lead times of 228h and 324h. But at lead time 132h, I see that the Atlantic influenced regions all do not perform very good. Has this something to do with the ocean/land interface? Could you investigate this and add a statement about this to your manuscript?

The method indeed does not seem to work well in the Atlantic influenced region, e.g., the west coast of the UK and the northwest tip of Spain. In these regions, the FWI is (similar to the mountainous regions) quite low, which can be seen in the newly added Figures of the FWI mean and 95th percentile in the supplement (new S2 and S3). This can be attributed to the increased rainfall caused by the ocean's influence in these regions. We added a comment about this in the description of Fig.6.

48. Line 234: please shorty reintroduce the term FWI analysis, e. g. FWI derived from the ensemble forecast (analysis). For people skimming your paper it might appear as you analyze the FWI, which you don't.

We agree that this would be beneficial for readers only skimming the paper and reintroduced the FWI analysis term as follows:

*"We used ECMWF high-resolution weather forecasts with the shortest possible lead time to calculate the FWI. The FWI values from these forecasts are referred to as the FWI analysis and are used as a substitute for observations. Although the FWI analysis seems to slightly underestimate the FWI, a good correlation is observed."*

49. Lines 236 -239: Please be consistent when using lead time units. It is hard to transfer from 84 hours to 8 to 10 days. You can state the lead time in hours and add the days in brackets after the hours as you stated in your response to the reviewers.

We added the days in brackets after the hours. It should now be easy to compare the short lead times to the 8 to 10 days mentioned later.

50. Line 240: this statement is weak and a reference to appendix (e.g. "see Fig S5-7") is missing. From these figures you can only read that the mean error is smaller in the summer months, but not how high or low the FWI is in the specific subregions. Therefore, you cannot state here that your findings are likely caused by low FWI values.

We added a reference to the appendix and also to the new supplement figures showing the mean and 95th percentile of FWI (S2,S3) to illustrate the range of FWI values in the subregions. From these figures, it becomes clear that in NEU the FWI is generally lower than in EUMED.

51. Lines 240: It is not clear what you consider high and low FWI values, please add values or define low or high FWI somewhere, e. g. "high FWI values (FWI > 1)".

We added a clarification what we mean with high and low FWI values.

52. Line 244: Rephrase "Although it would be ideal" to more scientific language.

We rephrased to "While it would be preferable" and hope it sounds more scientific.

53. Line 246: Define what you mean by "long-range" and "short-range" forecasts.

We changed the sentence referring to:

*"While forecasts of potential fire danger for extended periods (1-2 weeks) are valuable, short-term forecasts for the first 1-3 days, are usually more critical for firefighting resource management."*

In this way, we avoid calling it "long-range forecast" which is defined by the ECMWF as subseasonal (multiple months). However, in this context, we mean longer than short-term forecasts, e.g., 1-2 weeks. Short-term forecasts are forecasts up to 3 days which is stated in the following sub-clause.

54. Line 251: Move everything after "compared to more complex approaches" to the beginning of the discussion and conclusion chapter, e. g. to line 233 before you start with the sentence "We used…", to end your paper with this very good and strong statement: The improvement of FWI forecasts using the presented calibration method improves the ability to anticipate fire danger, ultimately supporting better response management and shows that a relatively simple method can provide good results compared to more complex approaches. In line 233 you can discuss that NGR is a good method in comparison to other methods (e.g. bias correction) for your research purpose.

We applied you suggested changes to the Discussion and conclusion chapter.